# Learning Neural Representations From Publicly Available Model Hubs

## Abstract

The weights of neural networks have emerged as a novel data modality, giving rise to the field of *weight space learning*. A central challenge in this area is that learning meaningful representations of weights typically requires large, carefully constructed collections of trained models, typically referred to as *model zoos*. These model zoos are often trained ad-hoc, requiring large computational resources, constraining the learned weight space representations in scale and flexibility. In this work, we drop this requirement by training a weight space learning backbone on arbitrary models downloaded from large, unstructured model repositories such as Hugging Face. Unlike curated model zoos, these repositories contain highly heterogeneous models: they vary in architecture and dataset, and are largely undocumented. To address the methodological challenges posed by such heterogeneity, we propose a new weight space backbone designed to handle unstructured model populations. We demonstrate that weight space representations trained on models from Hugging Face achieve strong performance, often outperforming backbones trained on laboratory-generated model zoos. Finally, we show that the diversity of the model weights in our training set allows our weight space model to generalize to unseen data modalities. By demonstrating that high-quality weight space representations can be learned *in the wild*, we show that curated model zoos are not indispensable, thereby overcoming a strong limitation currently faced by the weight space learning community. Code, pre-trained weights, and model collections can be found on `redacted`.

## 1 Introduction

Over the past years, weight space learning (WSL) has emerged as a vibrant research field casting neural-network parameters themselves as a data modality to learn representations from. WSL aims to learn representations of model weights given a population of models, i.e., a model zoo. Such learned representations can then be exploited for multiple downstream tasks: discriminative (e.g., predicting model properties such as accuracy directly from its weights (Unterthiner et al., 2020; Eilertsen et al., 2020; Martin et al., 2021; Schürholt et al., 2021; 2024; Navon et al., 2023; Zhou et al., 2023a)) or generative (generating new, unseen neural network weights for a given architecture and dataset (Schürholt et al., 2022b; Knyazev et al., 2023; 2024; Schürholt et al., 2024; Kofinas et al., 2023; Wang et al., 2024; 2025; Soro et al., 2024).

As promising as the exploitation of such learned representations for the above-mentioned downstream tasks is, previously proposed approaches in WSL (Unterthiner et al., 2020; Eilertsen et al., 2020; Schürholt et al., 2022b; Knyazev et al., 2023; Kofinas et al., 2023; Schürholt et al., 2024; Soro et al., 2024; 2025; Wang et al., 2024; 2025) share a common limitation: they require trained neural network models as input. Some methods are trained on multiple training checkpoints of a single model (Wang et al., 2024; 2025) while others use model zoos (i.e. populations of neural networks that are homogeneous in their training dataset and/or neural network architecture) to train the weight-space backbone (Schürholt et al., 2022a; 2024) or a combination of both (Soro et al., 2024).

The availability of homogeneous laboratory-trained model zoos therefore represents a major bottleneck: training them demands significant computational resources, particularly when scaling parameter count. And while some recent work (Schürholt et al., 2024; Kahana et al., 2024; 2025; Horwitz et al., 2024; 2025) suggests using publicly available models from repositories such as Hugging Face (HF),

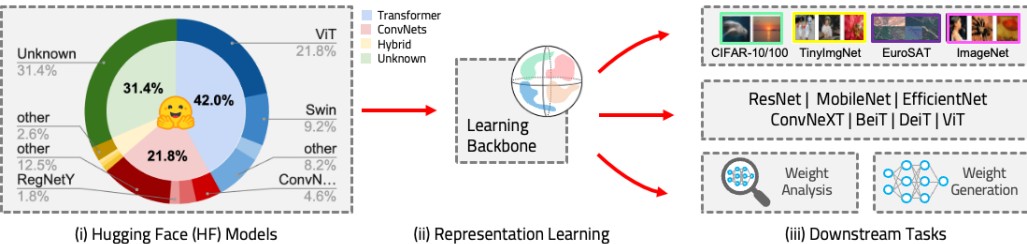

Figure 1: An overview of the proposed method. We train a weight space representation directly from the weights of downloaded models from Hugging Face. These models are, to a large extent, undocumented, trained on various datasets, and composed from different neural network architectures. Once a representation is learned from such a heterogeneous model collection, it can be exploited for multiple downstream tasks: either analyzing or generating model weights for multiple architectures and target datasets. Please note, all this is accomplished using the same single representation trained entirely from HF models.

the heterogeneity of neural network models on such platforms is beyond what current methods in WSL are able to train on.

To the best of our knowledge, only Horwitz et al. (2024) and Soro et al. (2024) leverage models taken from HF for weight space learning. Horwitz et al. (2024) group related models into model trees which can be leveraged for model retrieval and analysis; Soro et al. (2024) use individual models from HF in some experiments. In contrast, our work aims to design a method that can process and learn from *arbitrary* HF models that are heterogeneous in terms of architecture, training dataset and scale. Further, this work aims to train a single neural representation for all architecture and dataset combinations instead of one individual neural representations per architecture/dataset combination.

Indeed, using the weights of arbitrary models from HF as input for WSL is a non-trivial problem. The learning backbone should (i) handle models trained on different data distributions or tasks, (ii) be scalable to process larger models, and (iii) be able to process and embed different architectures. Additionally, a large fraction of these models are insufficiently documented (Horwitz et al., 2025).

In this work, we aim to close this gap and shift WSL from model zoo silos to the open, heterogeneous ecosystem of HF, home to over a million neural network models of different architectures and trained from various datasets. Learning weight space representations to capture the heterogeneity of such a diverse set of neural network models is challenging but potentially possible: Dravid et al. (2023) showed that different neural networks, composed from different architectures and trained for various vision tasks, share some common representations. Similar is hypothesized by Huh et al. (2024), where increased diversity during learning does lead to shared representation spaces.

Building upon an encoder-decoder transformer architecture (Schürholt et al., 2024; Soro et al., 2024; Wang et al., 2025), we propose the first WSL backbone whose training procedure is designed to be agnostic to: (i) model architecture, (ii) training dataset, (iii) model scale, and (iv) input modality for training. In Sec. 4, we demonstrate that the trained single representation can be used to generate more than 30 different architecture/dataset pairs ranging from ResNets (He et al., 2016b), ConvNeXts (Liu et al., 2022), EfficientNets (Tan & Le, 2019) to various transformer models (including ViT (Dosovitskiy et al., 2021), Swin (Liu et al., 2021b), BeiT (Bao et al., 2022), GPT-2 (Radford et al., 2018). This practically drops the aforementioned requirement to train on homogeneous model zoos for weight space representation learning. Summarizing, the contributions of this paper are: (i) training a single weight space representation from arbitrary models downloaded from HF, consisting of 171 billion individual weights, (ii) a novel backbone capable of learning a single task-agnostic and architecture-agnostic representation of weight spaces, (iii) generalizability in downstream task performance across different datasets and architectures.

## 2 HUGGING FACE MODEL COLLECTION

Learning representations of neural network models requires model zoos that are curated and trained in a controlled laboratory environment following a fixed training protocol to collect metadata during training, observe their learning progress, and fully document training trajectories from their start conditions until their converged performance. More formally, following Unterthiner et al. (2020), model zoos are defined as a collection of (converged) neural network models, each being configured by the tuple $\{\mathcal{D}, \lambda, \mathcal{A}\}$ with $\mathcal{D}$ being the dataset of samples, $\lambda$, the set of hyper-parameters used for training, (*e.g.*, loss function, optimizer, learning rate, weight initialization, batch-size, epochs), and $\mathcal{A}$, a specific neural network architecture.

Downloading neural network models from HF gives us a collection of models where $\{\mathcal{D}, \lambda, \mathcal{A}\}$ is potentially different for each model in the zoo, and in the worst case where $\mathcal{D}, \lambda$ is unknown due to a lack of documentation as reported by Horwitz et al. (2024; 2025). Learning a weight space representation from such heterogeneous or even unknown inputs is more challenging than training from laboratory-trained model zoos. One has to deal with unknown dataset distributions, unknown model performance, inconsistent model trajectories, and different architecture families or model trees.

**Download Protocol** To create the HF model collection used in this work, we exploit HF model tags, specifically including 'image-classification', 'image-segmentation', 'depth-estimation', and 'object-detection'. In total, when querying the HF API with these tags we have $22\,055$ models available, with $17\,011$ models for 'image-classification', $1\,381$ models for 'image-segmentation', $202$ models for 'depth-estimation', and $3\,461$ models for 'object-detection'. We constrained our experimental setup to computer vision models to be comparable to previous works. For each model in this set, we perform sanity checks: first, we verify if each model can be properly instantiated using the HF auto-classes for model loading, excluding models with missing weights, improperly saved checkpoints, or those requiring remote code execution for initialization. Once instantiated, we attempt to tokenize each model using the tokenization scheme discussed in Sec. 3.3. Successfully tokenized models are kept, and their model IDs are recorded for further processing. This procedure is done until subset of $2\,000$ training and $200$ validation models is retrieved.

**HF Model Collection** An overview of the composition of the retrieved HF model collection can be seen in Fig.1. Specifically, the included models fall under the families of Transformer (42.0%), ConvNet (21.8%) and hybrid (5%) architectures. Notably, a large percentage of these model architectures (31.4%) does not provide any information in the name and is classified as unknown. The composition of the collection is further analyzed into ResNets (He et al., 2016a), ConvNeXT (Liu et al., 2022), EfficientNet (Tan & Le, 2019), ViT (Dosovitskiy et al., 2021), SwinTransformer (Liu et al., 2021a), BeiT (Bao et al., 2022), DeiT (Touvron et al., 2021), and other architectures. Interestingly, half of the model collection appears to be trained on variations of ImageNet (Deng et al., 2009), whereas for the other half of the collection no information is provided. In the end, our HF model collection contains in total 171 billion individual parameters to be used for WSL.

## 3 METHODS

To accomplish training on uncurated models from HF, we require a learning backbone capable of scaling to arbitrary model sizes and capable of processing heterogeneous neural network architectures. Currently, no learning backbone would match these requirements (we discuss this in more detail in App. B). On the one hand, some existing learning backbones are able to scale but fail to process heterogeneous model architectures (Schürholt et al., 2024; Wang et al., 2024; 2025). SANE (Schürholt et al., 2024) requires homogeneous architectures of models in the zoo for the layer-wise loss normalization, p-diff (Wang et al., 2024), and RPG (Wang et al., 2025) require checkpoints saved during the training process of a single model. On the other hand, there are methods that can process heterogeneous architectures but are not able to scale at the same time. For instance, with D2NWG (Soro et al., 2024) a unified backbone can only be trained on classifier-heads and small models. Therefore, our method bridges this gap in the literature by training a single weight-space representation on arbitrary models at scale.

While differing in their specific implementation, all these works have something in common: they are based on an encoder-decoder learning backbone which we also build upon. Specifically, we base our work on the encoder-decoder transformer setup of SANE (Schürholt et al., 2024).

### 3.1 PRELIMINARY: THE SANE BACKBONE

SANE is an autoencoder where both the encoder and the decoder are symmetric transformers. Given some input NN weights $W = [\boldsymbol{W_1}, ..., \boldsymbol{W_l}]$ where the $\boldsymbol{W_i}$ are the weight matrices for the different layers, we first tokenize them as a sequence of tokens $\boldsymbol{T} = [\boldsymbol{t_1}, ..., \boldsymbol{t_n}]$ where $\boldsymbol{t_i} \in \mathbb{R}^{d_t}$ are the individual tokens (cf. Sec. 3.3 for more details about the tokenization). We then pass them through the encoder $g_\theta$ to embed them into a lower dimensional latent representation $g_\theta(\boldsymbol{T}) = \boldsymbol{Z}$, before the decoder $h_\psi$ reconstructs the tokens $h_\psi(\boldsymbol{Z}) = \hat{\boldsymbol{T}}$. To train this autoencoder, a combination of two losses is used. First, a contrastive loss (Chen et al., 2020) is used in the latent representation space, using augmentations such as permutations, noise and masking. Second, an MSE loss on $\boldsymbol{T}$ and its reconstruction $\hat{\boldsymbol{T}}$.

After training, the encoder can be used to embed unknown models into the latent representation $\boldsymbol{Z}$ for discriminative downstream tasks such as accuracy or hyperparameter prediction. Alternatively, by sampling new representations $\tilde{\boldsymbol{Z}}$ and passing them through the decoder, one can generate synthetic tokens $\tilde{\boldsymbol{T}} = h_\psi(\tilde{\boldsymbol{Z}})$, which can be detokenized into NN weights $\tilde{W}$. The difficulty lies in identifying the distribution $P(\boldsymbol{Z})$ of the latent representation space that can be decoded to a functional NN in the target domain. SANE uses Kernel Density Estimation (KDE) to model the target distribution based on one or multiple trained NNs (tokenized as $\boldsymbol{T_a}$) as *anchors* to sample $\tilde{\boldsymbol{Z}}$ in the vicinity of their latent representations $\boldsymbol{Z_a} = g_\theta(\boldsymbol{T_a})$. We discuss the different components in more detail in App. B.2.

To enable such an encoder-decoder learning backbone to learn a weight space representation from diverse architectures included in the HF model collection, significant modifications to the backbone are required. In the following, we outline these changes, including the design choices and implementation details.

### 3.2 MASKED LOSS NORMALIZATION (MLN)

Previous WSL work established that different weight distributions between different layers present a challenge for weight representation learning (Peebles et al., 2022; Schürholt et al., 2022a; 2024; Wang et al., 2025). As remedies, they propose to either normalize the weights per layer across the entire dataset as a preprocessing step, or normalize the loss contribution accordingly. Both approaches present challenges for large, inhomogeneous weight datasets. They are not immediately applicable for varying architectures since they compute normalizations per layer and thus require matching architectures. Further, such normalizations may fail for models trained on different computer vision datasets with different weight distributions. Normalizing the loss per layer inherits these constraints.

To tackle this challenge, we propose to normalize loss contributions *per-token* at runtime. This has two benefits: (i) it simplifies the normalization and operates across different model architectures and weight distributions, (ii) the representation learning model still operates in weight space, which simplifies evaluating weight generation.

We normalize each original token $\boldsymbol{t_i} \in \boldsymbol{T}$ and its predicted reconstruction $\widehat{\boldsymbol{t_i}}$ into $\boldsymbol{\tau_i}$ and $\widehat{\boldsymbol{\tau_i}}$ respectively:

$$\boldsymbol{\tau_i} = \frac{\boldsymbol{t_i} - \bar{t}}{\sigma_t}, \quad \widehat{\boldsymbol{\tau_i}} = \frac{\widehat{\boldsymbol{t_i}} - \bar{t}}{\sigma_t}, \tag{1}$$

where $\bar{t}$ and $\sigma_t$ are the mean and standard deviation of calculated over the current batch of tokens. Depending on the tokenization strategy used, tokens may includes zero-padding to harmonize token size, which can skew the mean and standard deviation estimators. We therefore ensure that both these estimators only take unmasked elements into account. When then compute the reconstruction mean-squared error loss between the normalized tokens $\boldsymbol{\tau_i}$ and $\widehat{\boldsymbol{\tau_i}}$. MLN is conceptually similar to normalization layers in neural networks. While normalization layers stabilize training by standardizing activations before they are passed forward, the goal of MLN is to stabilize weight-space representation learning by re-centering and rescaling tokens before their reconstruction error is computed. In both cases, normalization removes scale-related biases and ensures that optimization

focuses on the relative structure of the representation rather than raw magnitudes, enabling more robust learning across heterogeneous architectures.

### 3.3 Efficient Model Weight Processing

Below we outline our adaptations NN weight tokenization and encoding their structure to be processed by the WSL backbone, with a focus on processing diverse architectures and reducing memory overhead to enable training of a *single* weight space representation instead of training multiple representations for different settings.

**Tokenization**   To effectively train an autoencoder on neural network parameters, there are two main approaches followed in the literature. The works of Kofinas et al. (2023); Lim et al. (2024); Knyazev et al. (2024) represent neural networks as graphs and use graph neural networks to process them. Other approaches flatten the entire neural network into a 1D vector before processing (Schürholt et al., 2022a; Wang et al., 2024), with (Soro et al., 2025) additionally using a VQ-VAE model to generate discrete token representations.

Further splitting the flattened weights into tokens of a fixed size has been proposed in SANE (Schürholt et al., 2024) to address scaling issues with embedding larger architectures. SANE follows a tokenization approach in which the parameters of the model are divided into chunks that are later processed by an autoencoder. Given neural network weights $W = [W_1, \ldots, W_l]$, where $W_i$ denotes the weight matrix of the $i$-th layer, each $W_i$ lies in $\mathbb{R}^{c_{\text{out}} \times c_1 \times \cdots \times c_{\text{in}}}$ with $c$ representing the number of channels. Each $W_i$ is flattened into a 2D matrix $X_i \in \mathbb{R}^{c_{\text{out}} \times c_r}$, where $c_r = c_1 \cdots \cdots c_{\text{in}}$. The weights are then sliced row-wise, along the outgoing channel dimension, and each resulting vector is partitioned into tokens $t$ of length $d_t$. If $d_t \nmid c_r$, the final token is zero-padded to length $d_t$. The full token sequence is obtained as $T = [t_1, \ldots, t_n]$, constructed by stacking all tokens from all weight matrices in order.

Depending on the number of weights per channel in the individual layers, this often leads to very *sparse* tokens that include a significant portion of zero-pads. This is especially pronounced when tokenizing diverse architectures as the token size cannot be optimized to minimize the amount of padding required for that single architecture.

As these pads still take up space in memory and need to be processed, they represent an obstacle in scaling up the learning backbone to larger architectures and more diverse models. To address this limitation, we explore using a *dense* tokenization instead, similarly to what Wang et al. (2025) developed in parallel to our work. In that case, for every $W_i \in \mathbb{R}^{c_{\text{out}} \times c_1 \times \cdots \times c_{\text{in}}}$, we flatten it to $X_i^{\text{flat}} \in \mathbb{R}^{c_{\text{flat}}}$ with $c_{\text{flat}} = c_{\text{out}} * c_1 * \ldots * c_{\text{in}}$. We then cut the resulting vector $X_i^{\text{flat}}$ into tokens $t$ of length $d_t$. If $d_t \nmid c_{\text{flat}}$, we zero-pad the last token to length $d_t$, before concatenating all tokens for all layers in order to obtain $T = [t_1, \ldots, t_n]$. Given that we zero-pad per layer, and not per outgoing channel anymore, the amount of padding is much lower. In App. D.4.2, we explore the impact of dense and sparse tokenization on token sparsity, memory footprint and model performance.

**Sinusoidal Positional Encoding**   Because the backbone is based on the transformer architecture, positional encodings are required to represent the sequential structure of the input data. In our case, token positions are represented with a three-dimensional vector $P = [n, l, k]$, where $n$ indicates the position of the token in the full model sequence, $l$ corresponds to the layer index, and $k$ to the token position within the layer. SANE uses learned positional embeddings which in our case is not feasible given the diversity and scale of the downloaded HF dataset. Particularly, the number of parameters required for learned positional embeddings grows with the sequence length of the flattened weights, leading to a significant memory overhead when scaling to larger architectures included in our training set. The SANE backbone trained on ResNet-18 models contains ~865M trainable parameters out of which ~57M are used for the position embedding with a max dimension of $P = [55000, 100, 550]$. In our case the resulting embedding matrix would become significantly larger as the HF trainset contains models with up to 1.3B parameters compared to ~12M params of a ResNet-18 (see App. D.4.2 for more details). To solve this issue, we replace learned positional embeddings with sinusoidal positional encodings (Dosovitskiy et al., 2021) which provide a parameter-free approach for encoding position. This inherently scale-invariant method can efficiently support models of varying sizes, allowing the backbone to capture relative positions which are of utmost importance in our heterogeneous WSL context by exploiting the linear relationship between the sinusoidal positional encodings.

## 4 EXPERIMENTS

In this Section, we test our training pipeline using models from the HF model collection (Sec. 2) as training data. First, we assess the methodological adaptations described in Sec. 3. We then evaluate the proposed method as a whole, hypothesizing that with an increasing amount of training samples from HF the trained backbone is able to compensate potential noisiness in our HF model collection when compared to training with laboratory-trained model zoos. To that end, we directly compare our approach to SANE, but unlike SANE which trains individual weight space backbones for each architecture-dataset pair, we train a single backbone across HF models. We design our experiments to run on a single H100 GPU. Training hyperparameters and time as well as implementation details for all experiments are detailed in App. C.

We focus on generative downstream tasks and include discriminative results in App. D.2. For the evaluation of our generative capabilities we test the performance of generated models either without any updates of trainable parameters or with a few epochs of finetuning. Our interest lies in exploring whether our backbone is able to learn a lower dimensional representation from heterogeneous HF models. To that end we use the `subsampling` method introduced in (Schürholt et al., 2024) using tokenized NN models as anchor samples to fit the KDE as detailed in Sec. 3.1. Importantly, this approach does not aim to produce fully trained models directly from the decoder. Rather, we evaluate whether a single backbone trained on heterogeneous HF models can still learn a useful shared latent representation compared to model-zoo trained baselines. For larger architectures, where initial performance is very low we measure how effectively generated weights converge when finetuning the models. For the initial experiments, we use ResNet-18 models from the model-zoo dataset (Schürholt et al., 2022c) as anchors. To evaluate different architectures, we use models from the Timm library (Wightman, 2019) as anchors.

### 4.1 MASKED LOSS NORMALIZATION (MLN)

In the first experiment, we focus on validating the core methodological changes introduced in this work, aiming to assess their impact on representation quality, convergence stability, and general performance. We focus on the Masked Loss Normalization (MLN) and provide further ablations on the tokenization scheme and positional encodings in App. D.4. We evaluate whether the masked loss normalization allows the backbone to adequately capture the different weight distributions of different layers and models which has been shown to be problematic when not normalizing the weights layer wise before training (Schürholt et al., 2022a; Wang et al., 2025). This is crucial, since the encoder-decoder approach proposed as backbone operates directly in weight space, where skewed or squashed distributions - even of just individual layers - can have a catastrophic impact on the performance of generated models. To validate the proposed loss normalization, we train the backbone on ResNet-18 models from the model zoo dataset (Schürholt et al., 2022c) and generate models for three different datasets to compare the proposed masked loss normalization (MLN) to the baseline in isolation.

**Loss normalization at runtime allows training without global weight normalization during pre-processing** The experiment demonstrates that training with masked loss normalization is a suitable replacement for global weight normalization at dataset preprocessing time. Further, we did not encounter training instabilities, which might have been introduced for padding-heavy tokens. These experiments confirm transformer based encoder-decoder backbones such

Table 1: Accuracy of generated ResNet-18 models. We compare SANE with layer-wise loss normalization (LWLN) as baseline with the performance when training with the proposed masked loss normalization (MLN).

| Method | CIFAR10 | CIFAR100 | TIN |
|--------|---------|----------|-----|
| LWLN | **68.6±1.2** | 20.4±1.3 | 11.7±0.5 |
| MLN | 60.8±0.7 | **29.00±0.7** | **24.76±0.2** |

as SANE can be trained with the proposed masked loss normalization allowing training on arbitrary architectures given the same token size. Compared to the baseline (Tab. 1) our approach outperforms SANE on CIFAR100 and TinyImageNet, while showing slightly worse performance on CIFAR10. We include further analysis and discussion in App. D.4.

Table 2: Accuracy of generated ResNet-18 models when restricting the HF model collection to specific architecture groups compared to training on HF models in the category computer vision. Three additional subsets of the model collection including only specific architecture groups are created for backbone training. Num models refers to the number of models that are tokenized in the trainset (90% of available models) since we use 10% of the models for the validation set. Num tokens refers to the approximate number of tokens in the training dataset when using dense tokenization with tokensize 230.

| HF Models | Num Models | Num Tokens (~) | CIFAR10 | CIFAR100 | TIN |
|-----------|-----------|----------------|---------|----------|-----|
| ResNet | 360 | 35M | 10.68±0.49 | 1.09±0.11 | 0.58±0.05 |
| ConvNext | 306 | 100M | 21.66±1.95 | 14.95±0.81 | 9.12±0.17 |
| ViT | 2000 | 590M | 31.38±3.91 | **32.24±1.34** | 20.14±0.88 |
| All | 2000 | 740M | **40.71±1.91** | 26.23±0.35 | **21.45±0.51** |

## 4.2 IMPACT OF HF MODEL COLLECTION COMPOSITION

To gain insights into how the composition of the HF training set influences downstream performance and whether such a training set-up is feasible, we systematically vary the diversity and scale of the models included in the HF model collection. In particular, we compare training on broad heterogeneous collections against subsets restricted to specific architecture families. In addition to the main HF model collection (Sec. 2) we create three subsets of our HF model collection, each restricted to a single family of NN architectures: ResNet, ConvNext, and ViT.

The ResNet and ConvNeXt subsets of our HF model collection include 360 and 306 models respectively, as there are fewer models on HF including those specific keywords in the name. Only the vision transformer subset is filled to 2000 models. Please note that the number of models in the dataset does not directly translate to the number of tokens, as that is also dependent on the model size. Therefore we use the number of tokens as a proxy for dataset size.

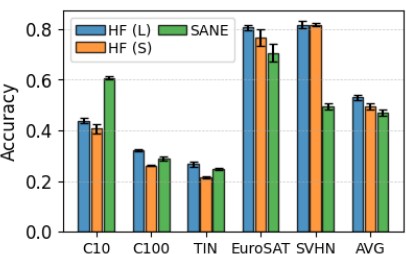

The results are included in Tab. 2 and show increasingly improved performance when including more models for training of the backbone even if they are not representative of the target architecture generated during inference. The performance of generated models when training the backbone on ResNets from HF is close to random guessing for CIFAR10, CIFAR100 and TinyImageNet. When training on HF ConvNext models, the performance of the generated methods is above random guessing across all datasets, but remains low. The backbone trained only on vision transformers is able to outperform the others even though we generate ResNet-18 models and not ViTs, simply because it includes much more samples. Training on the full vision dataset leads to a performance plateau: improving

Figure 2: Accuracy of generated ResNet-18 models on the respective target image datasets. No trainable parameters are updated before the performance evaluation. We compare training on homogeneous model zoos using SANE with MLN, to training the backbone on HF models. HF (S) and HF (L) designate the small and large versions of the backbone, respectively. With the exception of CIFAR-10, our approach outperforms model zoo training for all datasets.

on CIFAR10 and TinyImageNet while showing slightly worse performance on CIFAR100. We run further ablations on HF model collection size in App. D.4.

## 4.3 GENERATING WEIGHTS FOR VARYING DATASETS

For the next experiment, we include a broader evaluation across a variety of downstream datasets. In addition to CIFAR10 (C10), CIFAR100 (C100) (Krizhevsky & Hinton, 2009), and TinyImageNet (TIN) (Le & Yang, 2015), we generate model weights for SVHN (Netzer et al., 2011) and EuroSAT (Helber et al., 2019) while keeping the architecture fixed to a ResNet-18. We compare to SANE with MLN trained on CIFAR10 and CIFAR100 model zoos as baseline. Given the results from the previous experiment, we hypothesize that this is due to the vision dataset being too large for our backbone configuration and, therefore, run an additional experiment where we scale the backbone. Specifically,

we train a small variation (~450M params) as before and a large variation (~900M params) which is comparable to SANE (~865M params). The results (Fig. 2) show improved performance albeit at the cost of longer training time. Compared to the baseline, we observe lower performance on CIFAR10 but higher performance across all other tested datasets when generated with the large HF trained backbone. We analyze whether the increased training time when scaling the backbone is an acceptable tradeoff by generating a wider variety of architectures in the next experiment.

## 4.4 GENERATING WEIGHTS FOR VARYING ARCHITECTURES

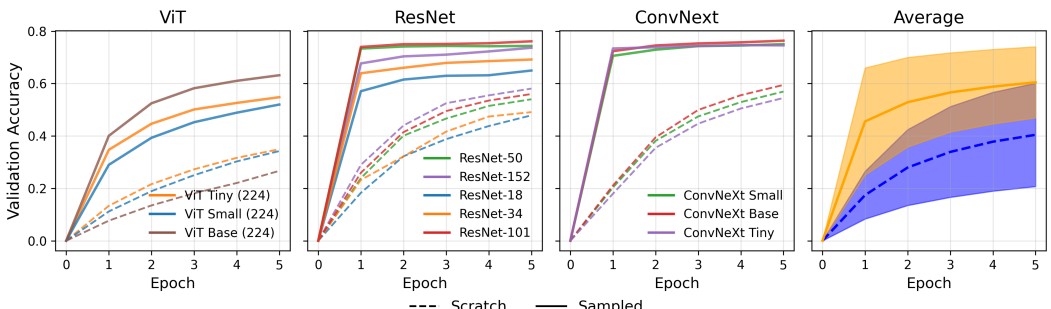

Figure 3: Accuracy of generated models on ImageNet-1K after a few epochs of finetuning. We show the performance of three different architecture types as well as the overall mean±std accuracy over all generated models comparing to training from scratch.

Having established that it is feasible to use HF models as training data, we are interested in the scalability of our proposed approach. In particular, we want to find out whether through the increased heterogeneity of our training dataset we are able to generate models for a wider variety of architectures compared to the baseline. For this, we reverse the generation procedure: we keep the downstream dataset fixed to ImageNet-1K and generate different architectures using the Timm library (Wightman, 2019) to instantiate models, which greatly simplifies the finetuning and evaluation pipeline. In total, we generate weights for 25 different architectures ranging from ResNets (He et al., 2016b), ConvNeXts (Liu et al., 2022), EfficientNets (Tan & Le, 2019) to various transformer models (including ViT (Dosovitskiy et al., 2021), Swin (Liu et al., 2021b), BeiT (Bao et al., 2022), GPT-2 (Radford et al., 2018)) and more. The full list of generated architectures is included in App. C including the hyperparameters used to finetune the models. In Fig. 3 we show the results for selected architecture types as well as the mean±std accuracy over all generated models compared to training the model with the exact same configuration from scratch. Full results for each architecture are included in App. D.1.

The results show that our backbone is able to generate a wide variety of architectures with significant and consistent performance gains over training from scratch. However, the benefit is only visible when finetuning the models: the initial performance remains slightly above or at random guessing. This is inline with previous findings (Knyazev et al., 2024; Meynent et al., 2025) showing that generated weights or learned weight updates can introduce small impurities that have a catastrophic impact on initial performance when scaling to larger architectures and more complex datsets. This can generally be fixed with a few optimization steps using standard SGD-based optimizers.

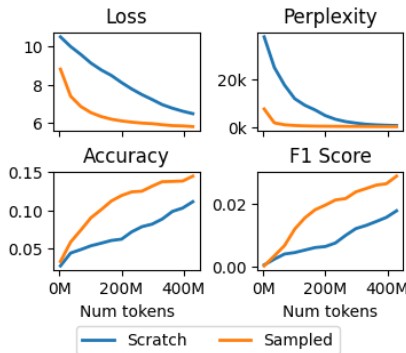

Figure 4: Comparing a generated GPT-2 model on OpenWebText to training from scratch. Results show the performance on the minival split and show that the backbone can process any type of model regardless of modality.

As baseline we train our backbone on ResNet-18 models from the CIFAR10 model zoo (Schürholt et al., 2022c). Here we include all proposed adaptations from Sec. 3 as otherwise scaling to larger architectures is not feasible. The results show that SANE remains competitive when the generated architecture is similar to the training data (e.g., ResNet or small ConvNeXt models), but struggles to scale to larger or more distinct architectures. Likewise, smaller backbones show diminishing

or negligible gains over training from scratch as model size increases. Full results are provided in App. D.3.

## 4.5 GENERATING WEIGHTS FOR ANOTHER MODALITY

To showcase the ability of the weight-space backbone to embed any architecture regardless of size and downstream modality, we further test whether our backbone that was trained on models tagged with computer vision tasks (Sec. 2) is able to generate weights for the initialization of a GPT-2 language model (~125M parameters) (Radford et al., 2018). We finetune the model on OpenWebText (Gokaslan et al., 2019) to assess initial trends. We show the validation loss, perplexity, accuracy, and F1 score of the generated model compared to training from scratch. Results (Fig. 4) show that our backbone is able to generate weights that converge faster compared to random initialization. This result demonstrates that our method can process and generate diverse model types even when they fall outside the type of models used to train the backbone. However, given that we do not restrict the model collection beyond basic feasibility checks and only filter based on (computer vision) tags, we cannot guarantee that the HF dataset does not include weights of models trained jointly with language embeddings (e.g., through CLIP (Radford et al., 2021)) and whether initialization benefits remain when training on a controlled zoo that contains only vision models of similar scale. Nevertheless, this highlights an exciting direction for future work: by scaling the backbone and expanding the HF training collection to encompass additional modalities and more models, it may become feasible to train a unified weight-space backbone spanning vision, language, and other modalities.

## 5 RELATED WORK

Recent advancements in representation learning for neural network weights have introduced various methods for analyzing and generating model weights. Hyper-Representations (Schürholt et al., 2021; 2022b;a) employ an encoder-decoder framework with contrastive guidance to learn weight representations for property prediction and model generation. Alternatively, some methods use diffusion on weights for generation purposes (Peebles et al., 2022; Wang et al., 2024; 2025; Soro et al., 2024). Graph-based methods (Zhang et al., 2019; Kofinas et al., 2023; Lim et al., 2024), Neural Functionals (Zhou et al., 2023a;b; 2024), and related approaches like Deep Weight Space (DWS) (Navon et al., 2023; Zhang et al., 2023) learn equivariant or invariant representations of weights. In conjunction with backbone architectures, data augmentations have been proposed to improve generalization of WSL methods (Schürholt et al., 2021; Shamsian et al., 2024). Other approaches focus on probing intermediate layers of neural network to predict attributes (Horwitz et al., 2024; Kahana et al., 2024; 2025). Also related are HyperNetworks, which directly use the underlying data and labels to guide the weights-generation signal (Ha et al., 2016; Knyazev et al., 2021; 2023; Brock et al., 2017; Navon et al., 2021). Our work focuses on the autoencoder-based approaches due to their broad applicability across various architectures and downstream tasks.

## 6 DISCUSSION

In this work, we have shown that training weight space models does not necessarily require laboratory-generated model zoos, which are expensive to train, take large amounts of storage and are therefore difficult to share with the larger community. Instead, we show the strong potential in training weight space models using publicly available model weights from repositories such as Hugging Face. This opens up multiple avenues for future work. First, our exploration has focused on a limited number of computer vision models. The training set of model weights from HF could be extended both in size and diversity by adding more models and covering more data modalities. Second, we have not applied manual curation beyond basic feasibility checks, meaning that the subset of models we train on may not fully represent the much larger pool of available weights. This raises the possibility of selection bias that could impact performance. While our goal is precisely to enable training on arbitrary, heterogeneous collections of models rather than carefully curated zoos, future work could further investigate how such biases may arise and to what extent they affect robustness. Finally, by leveraging models as diverse as those from HF, our work marks a significant step towards applying weight space models to downstream tasks, for which no model zoo is available.

## 7 CONCLUSION

Our work is, to the best of our knowledge, the first to overcome the need of model zoos, one of the limitations faced by the WSL community. To do so, we have first described how we collected models from the HF repository. We have then adapted an existing WSL backbone to work on our dataset, and have validated that our adaptations work and are good performance-efficiency trade-offs. We have then demonstrated though a variety of experiments that training a single autoencoder on HF data shows superior or similar levels of performance compared to SANE autoencoders trained on individual architecture-dataset pairs. Furthermore, we have shown that our single model is able to generate good initialization weights for a GPT-2 model demonstrating that the backbone is able to process models of varying architectures, trained on different datasets and different modalities, opening up exciting directions for future work. Through these results, we demonstrate that training on model weights found *in the wild* is as good as, but often better, than on laboratory-generated model zoos, and it is both more general and more efficient. This finding makes the training of weight space models more accessible, and opens up exciting new opportunities in the development of methods and applications in WSL.

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

## A TECHNICAL APPENDICES AND SUPPLEMENTARY MATERIAL

We structure the appendix as follows:

App. B provides more details about our reasoning for the backbone choice as well as a brief description of core methods of SANE that we use as basis for our method and approach.
App. C includes details about the hyperparameters and architecture of the backbones and baselines as well as the finetuning configuration used in the experiments.
App. D includes additional details and results for the baselines as well as further ablations for the method.

## B WEIGHT SPACE BACKBONES

### B.1 CHOICE OF THE WEIGHT SPACE BACKBONE

Table 3: Summary of the capabilities of different weight-space methods. [1]By data agnostic, we express that the weight-space method does not need the data used to train the model weights it trains on. [2]By architecture agnostic, we express that the weight-space method does not need any knowledge of the model architecture, only the weights. [3]We show which methods can be used to generate synthetic neural network weights.

| Learning Backbones | Dataset-agnostic[1] | Architecture-agnostic[2] | Generative capabilities[3] |
|---|---|---|---|
| Weight Statistics | ✓ | ✓ | ✗ |
| DWSNet | ✓ | ✗ | ✗ |
| Graph Methods | ✓ | ✗ | ✗ |
| Functionalist Methods | ✗ | ✓ | ~ |
| Diffusion Methods | ✓ | ✗ | ✓ |
| Hyper-Representations | ✓ | ~ | ✓ |

In this Section, we explore different weight space backbones from the literature and evaluate how fit they are for our use-case. In particular, they should be capable of handling heterogeneous, often poorly documented (Horwitz et al., 2025) models from the HF repository Hugging Face, Inc. (2025). This is critical since we cannot directly filter models based on performance or other attributes unless we constrain our dataset creation to the smaller subset of labeled models. We summarize the capabilities of existing weight-space backbones in Tab. 3. There, we show that most of the existing weight-space backbones come with inherent limitations. Simple weight statistics have been shown to be very cheap and effective for discriminative downstream tasks such as predicting model performance and hyperparameters, they can be used in a way that is architecture agnostic, but they are not adapted for neural network weights generation. DWSNet (Navon et al., 2021) and graph-based backbones (Kofinas et al., 2023; Lim et al., 2024; Knyazev et al., 2024) use different approaches to encode the inherent symmetries and structures of neural networks, making them by definition dependent on the underlying models' architecture. Functionalist methods (Herrmann et al., 2024; Meynent et al., 2025) that focus on the models' outputs rather than their weights can easily be made architecture agnostic but rely on relevant data samples for probing and are, therefore, not data agnostic. Existing diffusion-based methods (Soro et al., 2024; Wang et al., 2024) for neural network weight generation are currently not completely architecture agnostic. More specifically, both the diffusion model and the learning backbone are usually trained for a single network architecture at a time, in particular when scaling to larger architectures. This limitation also prevents these approaches from being fully dataset- and task-agnostic, since the architecture is usually tied to both the task and the dataset and often requires additional labeled data for conditioning that is not available for most HF models. Finally, the SANE backbone (Schürholt et al., 2024), while not entirely architecture agnostic, can be adapted to be with some targeted changes; it can also perform neural network weights generation. For these reasons, we base our work on the SANE architecture.

### B.2 SANE

In this section we provide additional details about the Sequential Autoencoder for Neural Embeddings (SANE) (Schürholt et al., 2024) method and approach.

### B.2.1 Pre-Processing and Tokenization

Before training the model-zoos need to be sliced into tokens to allow scaling the hyper-representation to larger model-zoos. The required steps are described below.

**Layer wise loss normalization** Schürholt et al. (2022b) mention that due to the reconstruction loss being based on a mean squared error, the hyper-representation will learn to reconstruct the weights to evenly spread the reconstruction error across all weights and layers in the weight vector ($w$). However, the distribution of weight magnitudes can differ significantly between layers and also between different models even if they share the same architecture. This can lead to an issue where layers with large-magnitude weights and broader distributions are reconstructed accurately, but those with smaller magnitudes and narrow distributions can be neglected. This imbalance can make these smaller-weight layers a weak point in the reconstructed models, which can lead to a severe drop in performance, sometimes as low as random guessing Schürholt et al. (2022b; 2024). They therefore propose to normalize the weights of the input model-zoo as a first step. The weights are normalized with the mean ($\mu_l$) and standard deviation ($\sigma_l$) calculated per layer throughout the whole model zoo. However, as outlined in Sec. 3 this is not feasible when training on diverse architectures with varying depth and width without significant engineering overhead and why we propose the masked loss normalization at runtime instead which greatly simplifies pre-processing and evaluation while achieving comparable results.

**Tokenization** The weights are reshaped into 2D matrices and sliced row-wise along the outgoing channels (which we refer to in the paper as *sparse* tokenization). These slices are divided into multiple parts based on a predefined token size ($d_t$). If a slice does not fill an entire token, zero padding is applied to reach the required token size. Each token is then augmented with a 3-dimensional positional embedding, as explained in Sec. 3.3.

**Windowing** From the complete sequence of tokens and their positional embeddings, a random subset is selected, consisting of $n$ consecutive tokens up to a specified window size ($ws$). The model is trained using these windows, enabling it to handle large models by focusing on manageable segments at a time. During training, one window is sampled per model, ensuring that a batch contains tokens from different models. This also allows to draw different windows at each training iteration to further mitigate the risk of over-fitting. By decoupling the computational requirements from the input model size, SANE can scale to any architecture, regardless of its size. Through the tokenization process, it is also possible to train on different input architectures simultaneously as long as they are preprocessed and sliced into the same token and window size. For our experiments we preprocess the models into windows of tokens of length 4096 out of which a random subset of 512 tokens is sampled during each training iteration to reduce the risk of overfitting and improving training time.

**Augmentations** The original SANE implementation uses three different data augmentations in the context of its contrastive loss: masking, additive white gaussian noise and permutations. For the latter, the authors leverage the existing symmetries in neural networks that make it possible to swap the order of neurons without changing the input-output function of the model Ainsworth et al. (2022); Hecht-Nielsen (1990). When using model weights from HF that are not properly documented, it is not possible to know which permutations conserve this input-output function without knowledge about the architecture. For this reason, we do not explicitly deal with the symmetries of the models in our dataset, and only use the masking and noising augmentations to investigate whether it is still possible to achieve comparable performance to previous work.

### B.2.2 Downstream Tasks

SANE is a unified model and approach for discriminative and generative downstream tasks. The following paragraphs briefly introduce the methods used for both task families.

**Discriminative Tasks** For discriminative downstream tasks, one can pass a set of tokens $\boldsymbol{T}$ representing a model's weights through the SANE encoder $g_\theta$ to obtain a latent representation $g_\theta(\boldsymbol{T}) = \boldsymbol{Z}$. The latent representations is then averaged over all tokens to a single vector $\bar{\boldsymbol{Z}} \in \mathbb{R}^{d_t}$, which is processed with a linear probe or a Multi Layer Perceptron (MLP) to predict model properties. These

include as accuracy, loss, generalization gap, training epoch, model-zoo, etc. The predictive performance of the linear probe/MLP can be measured in terms of explained variance ($R^2$) on the test set in the case of a regression (e.g., accuracy with continuous values) or in terms of accuracy in the case of classification (e.g., prediction of the architecture of the embedded model).

**Generative Sampling of Models**   SANE is also able to sample models directly out of the representation space. The difficulty lies in identifying the distribution $P(\boldsymbol{Z})$ of the latent representation space that can be decoded to a functional NN in the target domain. SANE uses a Kernel Density Estimation (KDE) to model the target distribution $P(\boldsymbol{Z})$ by tokenizing a few *anchors* $\boldsymbol{T_a}$ and projecting them to the latent space using the encoder $\boldsymbol{Z_a} = g_\theta(\boldsymbol{T_a})$. The resulting KDE over the $\boldsymbol{Z_a}$ is then broadly sampled to identify regions in the distribution with high probability of the desired target properties. The sampled representations $\tilde{\boldsymbol{Z}}$ are passed through the decoder $h_\psi$ to generate synthetic tokens $\tilde{\boldsymbol{T}} = h_\psi(\tilde{\boldsymbol{Z}})$, which can then be detokenized into neural network weights $\tilde{W}$.

**Batch Norm Conditioning**   Larger models often include batch norm layers that in part contain parameters that are not trained via backpropagation and are only updated during the forward pass. Since the distribution of these weights differs significantly from trainable parameters, they are excluded from sampling. Instead, batch norm conditioning is performed, which updates (only non-trainable) parameters during a few forward passes on the target dataset to align these parameters with the trainable weights before evaluating the accuracy Schürholt et al. (2024).

**Haloing**   Optionally, SANE allows for the use of haloing, where context weights are added before and after the window before processing by the encoder. This added context is processed normally but disregarded after reconstruction by the decoder.

## C   IMPLEMENTATION DETAILS

Table 4: Implementation Details for Hugging Face Training and Baseline

| Hyper-Parameter | HF-Small | HF-Large | SANE (MLN) |
|---|---|---|---|
| tokensize (sparse) | 288 | - | 288 |
| tokensize (dense) | 230 | 230 | - |
| loss norm | MLN | MLN | MLN |
| pos embed | sinusoidal | sinusoidal | learned |
| window size | 512 | 512 | 256 |
| model dim | 1536 | 1536 | 2048 |
| latent dim | 128 | 128 | 128 |
| num transformer layers | 8 | 16 | 8 |
| num transformer heads | 8 | 8 | 8 |
| learning rate | $2e-5$ | $2e-5$ | $2e-5$ |
| weight decay | $3e-9$ | $3e-9$ | $3e-9$ |
| scheduler | OnceCycleLR | OnceCycleLR | OnceCycleLR |
| num training epochs | 100, 300 | 300 | 60 |
| batch size | 64 | 64 | 32 |
| gradient accumulation steps | - | 2 | - |
| num params (~) | 456M | 900M | 865M |
| training time | ~54h, 144h | ~198h | ~20h |
| training dataset | HF | HF | Model-Zoos |

In Tab. 4, we provide additional information on the training hyper-parameters and architecture configuration of the two HF trained backbones (HF-Small and HF-Large) as well as for testing our masked loss normalization (MLN) on model-zoo data (SANE MLN). We keep the baseline experimental set-up to validate the MLN as close as possible to the original SANE implementation in order for the results to be comparable. For training on HF data we introduce additional changes to the model size, number of training epochs and number of layers to balance the trade-off between performance and efficiency. We train the small backbone (HF-Small) for 100 epochs for comparing dense and sparse tokenization as well as for the ablations (App. D.4) to reduce training time. For comparing to the large backbone (HF-Large) we also train the small backbone for 300 epochs

(App. D.1) for evaluating across different architectures. The SANE baseline evaluated on ImageNet-1K is trained according to the HF-Small configuration for 100 epochs as we observed degrading performance when training for longer on homogeneous model zoos.

## C.1 LIST OF GENERATED MODELS

For our experiments across different datasets we use ResNet-18 models from publicly available model zoo datasets (Schürholt et al., 2022c; Honegger et al., 2023) with the exception of SVHN where we train 5 models from scratch. Below we list all generated architectures from the Timm (Wightman, 2019) library that we finetune on ImageNet-1K. We organize the models in architecture groups and specify the sampled architectures within that group. The specific instance of the timm model used is specified within parentheses.

- ConvNext
    - Tiny (convnext_tiny.fb_in1k)
    - Small (convnext_small.fb_in1k)
    - Base (convnext_base.fb_in1k)
- ResNet
    - ResNet-18 (resnet18)
    - ResNet-34 (resnet34)
    - ResNet-50 (resnet50)
    - ResNet-101 (resnet101)
    - ResNet-152 (resnet152)
- DenseNet
    - DenseNet 121 (densenet121.ra_in1k)
- EfficientNet
    - EfficientNet V2 Small (tf_efficientnetv2_s.in1k)
    - EfficientNet V2 Medium (tf_efficientnetv2_m.in1k)
- MobileNet
    - MobileNet V3 Small 100 (tf_mobilenetv3_small_)
    - MobileNet V3 Small 075 (tf_mobilenetv3_small_075.in1k)
    - MobileNet V2 100 (mobilenetv2_100.ra_in1k)
- Vision Transformer
    - Tiny-ViT (5M) (tiny_vit_5m_224.in1k)
    - Tiny-ViT (11M) (tiny_vit_11m_224.in1k)
    - ViT-T-16-224 (6M) (vit_tiny_patch16_224.augreg_in21k_ft_in1k)
    - ViT-S-16-224 (vit_small_patch16_224.augreg_in1k)
    - ViT-B-16-224 (vit_base_patch16_224.orig_in21k_ft_in1k)
- DeiT
    - DeiT-3-Base-16-224 (deit3_base_patch16_224.fb_in1k)
    - DeiT-3-Medium-16-224 (deit3_medium_patch16_224.fb_in1k)
- BeiT
    - BeiT V2 Base (beitv2_base_patch16_224.in1k_ft_in1k)
- Swin
    - Swin S3 Tiny (swin_s3_tiny_224.ms_in1k)
    - Swin S3 Small (swin_s3_small_224.ms_in1k)
    - Swin S3 Base (swin_s3_base_224.ms_in1k)

## C.2 Finetuning params

We use the same training pipeline for all finetuned models and only vary the batch size for the larger models (256 instead of 512). For generating weights we use haloing with an added context of 2*64 tokens per window (of size 512) and batch-norm conditioning. For our experiment on ImageNet we use one anchor sample from Timm (Wightman, 2019) and sample five models per architecture and finetune the best one. For sampling different datasets we use 5 anchor samples per target dataset and sample 50 models and keep the top 10. For data augmentations we use a random resized crop and random horizontal flip and normalize with the ImageNet mean and standard deviation. For validation we use rescaling with a center crop as well as normalization. We finetune the models for 5 epochs using the Adam (Kingma, 2014) optimizer with learning rate $1e - 3$ with autocast and gradient scaling. For sampling across different datasets we show the performance of sampled models without any finetuning of trainable parameters using the same augmentations as in the respective model zoo. The configuration and finetuning hyperparameters for the GPT-2 model are summarized in Tab. 5.

Table 5: Hyperparameters for GPT-2 finetuning

| Hyper-Parameter | Value |
| --- | --- |
| blocksize | 1024 |
| vocab size | 50'304 |
| num layers | 12 |
| num attention heads | 12 |
| embed dim | 768 |
| optimizer | AdamW |
| max learning rate | $6e - 4$ |
| weight decay | $1e - 1$ |
| scheduler | OnceCycleLR (Smith & Topin, 2018) |
| batch size | 64 |
| num training steps per iteration | 50 |
| num validation steps per iteration | 200 |
| evaluation frequency | 10 |
| gradient accumulation steps | 8 |
| dataset | OpenWebText |

# D  ADDITIONAL RESULTS & ABLATIONS

In this section we include detailed results for our generative (App. D.1) and discriminative (App. D.2) downstream tasks. We also include full results for the baselines in App. D.3.

## D.1  GENERATIVE RESULTS

**Full results per architecture**  In Tab. 6 we report the mean±std performance of all sampled models after 1-5 epochs of finetuning. As baselines, we include SANE trained on CIFAR10 with loss normalization and sinusoidal positional encodings, as well as models trained from scratch. Further below we show the individual results per sampled architecture after 1-5 epochs of finetuning comparing to training from scratch. Specifically, we show the performance of the large backbone in Tab. 7 and the performance of the small backbone in Tab. 8. The results of our baseline per architecture are included in Tab. 11. Furthermore, we show performance of sampled models after a fixed number of optimization steps in Tab. 9.

Table 6: Mean±std performance of generated models per backbone in percent after 1-5 epochs of finetuning. The results are calculated over all sampled models and show that training on HF models outperforms previous work trained on homogenous model zoos when scaling to larger and more diverse architectures. While the baseline achieves competitive results for architectures that are close to the training set (i.e. ResNets or ConvNexts) performance drops for other architectures and often leads to worse performance than training from scratch. Conversely, our HF trained backbones are able to consistently outperform training from scratch. The large backbone improves upon the small backbone in particular for larger sampled architectures.

| Backbone | Epoch | | | | |
|---|---|---|---|---|---|
| | 1 | 2 | 3 | 4 | 5 |
| Scratch | 17.43±9.28 | 27.97±14.77 | 33.91±17.66 | 37.82±19.24 | 40.43±20.12 |
| SANE | 17.36±18.94 | 24.32±23.01 | 28.33±24.90 | 30.85±25.94 | 30.72±26.96 |
| HF (Small) | 39.76±22.38 | 47.46±20.66 | 51.47±19.97 | 53.98±19.59 | 55.83±19.44 |
| HF (Large) | **45.55±20.84** | **52.88±17.44** | **56.61±15.41** | **58.81±14.49** | **60.48±13.85** |

Table 7: Performance of individual generated models after finetuning for 1-5 epochs on ImageNet-1K when sampling from the large HF-backbone (Tab. 4) vs. training from scratch. The HF backbone is able to outperform the small backbone and baselines in particular for larger architectures such as a Swin transformer or ResNet-152.

| Architecture Group | Architecture Name | Init | Epoch | | | | |
|---|---|---|---|---|---|---|---|
| | | | 1 | 2 | 3 | 4 | 5 |
| BEiT | Beitv2 Base Patch 16 | Sampled | **19.38** | **34.52** | **43.03** | **48.66** | **52.27** |
| | Beitv2 Base Patch 16 | Scratch | 6.87 | 9.94 | 9.48 | 10.98 | 12.24 |
| ConvNeXt | Convnext Base | Sampled | **72.35** | **74.52** | **75.31** | **75.72** | **76.36** |
| | Convnext Base | Scratch | 21.11 | 39.46 | 49.94 | 55.55 | 59.47 |
| | Convnext Small | Sampled | **70.57** | **72.91** | **74.28** | **74.49** | **75.04** |
| | Convnext Small | Scratch | 20.34 | 38.14 | 47.44 | 52.92 | 56.94 |
| | Convnext Tiny | Sampled | **73.38** | **73.87** | **74.31** | **74.68** | **74.55** |
| | Convnext Tiny | Scratch | 17.97 | 35.48 | 44.70 | 50.48 | 54.49 |
| DeiT | Deit3 Base Patch 16 | Sampled | **27.53** | **39.37** | **45.49** | **50.05** | **53.46** |
| | Deit3 Base Patch 16 | Scratch | 8.56 | 11.97 | 14.02 | 15.23 | 15.99 |
| | Deit3 Medium Patch 16 | Sampled | **37.09** | **48.15** | **53.85** | **56.78** | **59.61** |
| | Deit3 Medium Patch 16 | Scratch | 14.50 | 21.07 | 27.22 | 31.34 | 34.95 |

Table 7: Performance of individual sampled models after finetuning for 1-5 epochs on ImageNet-1K when sampling from the large HF-backbone (continued)

| Architecture Group | Architecture Name | Init | Epoch | | | | |
|---|---|---|---|---|---|---|---|
| | | | 1 | 2 | 3 | 4 | 5 |
| DenseNet | Densenet121 | Sampled | **56.75** | **61.86** | **64.37** | **65.34** | **66.03** |
| | Densenet121 | Scratch | 28.49 | 41.57 | 49.32 | 53.75 | 55.79 |
| EfficientNet | Efficientnetv2 M | Sampled | **58.19** | **65.51** | **67.82** | **70.35** | **71.37** |
| | Efficientnetv2 M | Scratch | 25.88 | 41.20 | 47.77 | 54.04 | 57.59 |
| | Efficientnetv2 S | Sampled | **64.29** | **69.12** | **71.13** | **72.80** | **73.39** |
| | Efficientnetv2 S | Scratch | 28.84 | 42.86 | 51.55 | 55.83 | 59.18 |
| MobileNet | Mobilenetv2 100 | Sampled | **37.56** | **47.78** | **51.95** | **54.74** | **56.64** |
| | Mobilenetv2 100 | Scratch | 18.32 | 30.13 | 37.92 | 43.14 | 46.26 |
| | Mobilenetv3 Small 075 | Sampled | **28.56** | **37.35** | **41.91** | **44.46** | **46.58** |
| | Mobilenetv3 Small 075 | Scratch | 18.42 | 28.06 | 33.59 | 37.28 | 40.28 |
| | Mobilenetv3 Small 100 | Sampled | **29.74** | **38.80** | **43.55** | **46.88** | **48.98** |
| | Mobilenetv3 Small 100 | Scratch | 19.89 | 29.29 | 35.75 | 40.15 | 43.14 |
| ResNet | Resnet101 | Sampled | **73.97** | **75.02** | **75.11** | **75.38** | **76.11** |
| | Resnet101 | Scratch | 25.96 | 41.45 | 49.49 | 53.51 | 56.00 |
| | Resnet152 | Sampled | **67.67** | **70.36** | **71.02** | **72.25** | **73.64** |
| | Resnet152 | Scratch | 28.99 | 44.01 | 52.48 | 55.49 | 58.07 |
| | Resnet18 | Sampled | **57.09** | **61.51** | **62.96** | **63.15** | **64.96** |
| | Resnet18 | Scratch | 18.30 | 32.25 | 38.77 | 43.78 | 47.86 |
| | Resnet34 | Sampled | **63.92** | **66.02** | **67.88** | **68.55** | **69.16** |
| | Resnet34 | Scratch | 23.24 | 32.18 | 41.65 | 47.41 | 49.10 |
| | Resnet50 | Sampled | **73.33** | **74.13** | **74.34** | **74.24** | **74.35** |
| | Resnet50 | Scratch | 23.92 | 40.10 | 46.63 | 51.45 | 54.00 |
| Swin | Swin S3 Base 224 | Sampled | **12.03** | **20.33** | **26.42** | **29.13** | **28.24** |
| | Swin S3 Base 224 | Scratch | 0.20 | 0.10 | 0.10 | 0.10 | 0.10 |
| | Swin S3 Small 224 | Sampled | **9.94** | **16.71** | **20.96** | **21.73** | **25.43** |
| | Swin S3 Small 224 | Scratch | 0.10 | 0.10 | 0.10 | 0.10 | 0.10 |
| | Swin S3 Tiny 224 | Sampled | **22.10** | **34.38** | **42.03** | **47.72** | **51.95** |
| | Swin S3 Tiny 224 | Scratch | 0.10 | 0.10 | 0.10 | 0.10 | 0.10 |
| ViT | Tiny Vit 11M 224 | Sampled | **42.25** | **54.31** | **58.48** | **61.89** | **63.07** |
| | Tiny Vit 11M 224 | Scratch | 24.42 | 42.43 | 49.50 | 54.88 | 57.24 |
| | Tiny Vit 5M 224 | Sampled | **37.16** | **48.95** | **55.43** | **58.73** | **60.87** |
| | Tiny Vit 5M 224 | Scratch | 29.08 | 43.20 | 49.88 | 53.96 | 56.12 |
| | Vit Base Patch 16 | Sampled | **40.05** | **52.45** | **58.20** | **61.03** | **63.17** |
| | Vit Base Patch 16 | Scratch | 7.67 | 13.49 | 18.14 | 22.04 | 26.68 |
| | Vit Small Patch 16 | Sampled | **28.98** | **39.30** | **45.24** | **48.88** | **51.98** |
| | Vit Small Patch 16 | Scratch | 11.20 | 18.94 | 25.03 | 30.27 | 34.23 |
| | Vit Tiny Patch 16 | Sampled | **34.72** | **44.65** | **50.10** | **52.60** | **54.80** |
| | Vit Tiny Patch 16 | Scratch | 13.34 | 21.61 | 27.27 | 31.65 | 34.93 |
| Mean (Sampled) | | | **45.55** | **52.88** | **56.61** | **58.81** | **60.48** |
| Mean (Scratch) | | | 17.43 | 27.97 | 33.91 | 37.82 | 40.43 |

Table 8: Performance of individual generated models after finetuning for 1-5 epochs on ImageNet-1K when sampling from the small HF-backbone (Tab. 4) vs. training from scratch. The small HF backbone achieves similar performance compared to the large variation on smaller models but fails to scale to larger architectures.

| Architecture Group | Architecture Name | Init | Epoch | | | | |
|---|---|---|---|---|---|---|---|
| | | | 1 | 2 | 3 | 4 | 5 |
| BEiT | Beitv2 Base Patch 16. Ft | Sampled | **16.06** | **29.46** | **38.07** | **45.24** | **50.41** |
| | Beitv2 Base Patch 16 | Scratch | 6.87 | 9.94 | 9.48 | 10.98 | 12.24 |
| ConvNeXt | Convnext Base | Sampled | **72.69** | **74.59** | **75.00** | **75.91** | **76.20** |
| | Convnext Base | Scratch | 21.11 | 39.46 | 49.94 | 55.55 | 59.47 |
| | Convnext Small | Sampled | **71.16** | **73.14** | **74.27** | **74.99** | **75.11** |
| | Convnext Small | Scratch | 20.34 | 38.14 | 47.44 | 52.92 | 56.94 |
| | Convnext Tiny | Sampled | **73.37** | **74.12** | **74.34** | **74.66** | **74.43** |
| | Convnext Tiny | Scratch | 17.97 | 35.48 | 44.70 | 50.48 | 54.49 |
| DeiT | Deit3 Base Patch 16 | Sampled | **22.00** | **32.45** | **39.16** | **44.04** | **47.75** |
| | Deit3 Base Patch 16 | Scratch | 8.56 | 11.97 | 14.02 | 15.23 | 15.99 |
| | Deit3 Medium Patch 16 | Sampled | **24.23** | **35.80** | **42.01** | **46.44** | **50.04** |
| | Deit3 Medium Patch 16 | Scratch | 14.50 | 21.07 | 27.22 | 31.34 | 34.95 |
| DenseNet | Densenet121.Ra In1K | Sampled | **50.07** | **56.85** | **60.08** | **62.33** | **63.46** |
| | Densenet121.Ra In1K | Scratch | 28.96 | 43.28 | 49.67 | 54.34 | 56.30 |
| EfficientNet | Efficientnetv2 M | Sampled | **57.14** | **64.68** | **67.54** | **69.62** | **70.56** |
| | Efficientnetv2 M | Scratch | 25.88 | 41.20 | 47.77 | 54.04 | 57.59 |
| | Efficientnetv2 S | Sampled | **65.77** | **70.25** | **71.93** | **73.02** | **73.84** |
| | Efficientnetv2 S | Scratch | 28.84 | 42.86 | 51.55 | 55.83 | 59.18 |
| MobileNet | Mobilenetv2 100.Ra | Sampled | **39.84** | **48.55** | **53.26** | **55.37** | **57.61** |
| | Mobilenetv2 100 | Scratch | 18.32 | 30.13 | 37.92 | 43.14 | 46.26 |
| | Mobilenetv3 Small 075 | Sampled | **26.48** | **35.52** | **39.95** | **43.12** | **45.41** |
| | Mobilenetv3 Small 075 | Scratch | 18.42 | 28.06 | 33.59 | 37.28 | 40.28 |
| | Mobilenetv3 Small 100 | Sampled | **29.88** | **38.23** | **43.04** | **46.21** | **48.09** |
| | Mobilenetv3 Small 100 | Scratch | 19.89 | 29.29 | 35.75 | 40.15 | 43.14 |
| ResNet | Resnet101 | Sampled | 21.48 | 40.03 | **49.66** | **54.92** | **58.37** |
| | Resnet101 | Scratch | **25.96** | **41.45** | 49.49 | 53.51 | 56.00 |
| | Resnet152 | Sampled | **33.88** | **47.94** | **54.90** | **56.05** | **59.88** |
| | Resnet152 | Scratch | 28.99 | 44.01 | 52.48 | 55.49 | 58.07 |
| | Resnet18 | Sampled | **57.50** | **62.06** | **63.04** | **64.32** | **65.27** |
| | Resnet18 | Scratch | 18.30 | 32.25 | 38.77 | 43.78 | 47.86 |
| | Resnet34 | Sampled | **63.82** | **66.80** | **67.94** | **68.75** | **68.38** |
| | Resnet34 | Scratch | 23.24 | 32.18 | 41.65 | 47.41 | 49.10 |
| | Resnet50 | Sampled | **72.39** | **72.68** | **73.74** | **72.44** | **74.50** |
| | Resnet50 | Scratch | 23.92 | 40.10 | 46.63 | 51.45 | 54.00 |
| Swin | Swin S3 Base | Sampled | **0.46** | **0.43** | **0.43** | **0.43** | **0.43** |
| | Swin S3 Base | Scratch | 0.10 | 0.10 | 0.10 | 0.10 | 0.10 |
| | Swin S3 Small | Sampled | 0.10 | 0.10 | 0.10 | 0.10 | 0.10 |
| | Swin S3 Small | Scratch | 0.10 | 0.10 | 0.10 | 0.10 | 0.10 |
| | Swin S3 Tiny | Sampled | **13.94** | **27.35** | **36.15** | **42.79** | **47.65** |
| | Swin S3 Tiny | Scratch | 0.10 | 0.10 | 0.10 | 0.10 | 0.10 |

Table 8: Performance of individual generated models after finetuning for 1-5 epochs on ImageNet-1K when sampling from the small HF-backbone (continued)

| Architecture Group | Architecture Name | Init | Epoch 1 | 2 | 3 | 4 | 5 |
|---|---|---|---|---|---|---|---|
| ViT | Tiny Vit 11M 224 | Sampled | **43.50** | **53.78** | **58.95** | **62.36** | **63.44** |
| | Tiny Vit 11M 224 | Scratch | 24.42 | 42.43 | 49.50 | 54.88 | 57.24 |
| | Tiny Vit 5M 224 | Sampled | **39.28** | **49.73** | **55.61** | **58.67** | **59.67** |
| | Tiny Vit 5M 224 | Scratch | 29.08 | 43.20 | 49.88 | 53.96 | 56.12 |
| | Vit Base Patch 16 224 | Sampled | **40.19** | **52.23** | **57.65** | **60.94** | **63.08** |
| | Vit Base Patch 16 | Scratch | 7.67 | 13.49 | 18.14 | 22.04 | 26.68 |
| | Vit Small Patch 16 | Sampled | **29.64** | **40.99** | **46.10** | **49.75** | **52.41** |
| | Vit Small Patch 16 | Scratch | 11.20 | 18.94 | 25.03 | 30.27 | 34.23 |
| | Vit Tiny Patch16 224 | Sampled | **29.17** | **38.67** | **43.81** | **47.04** | **49.59** |
| | Vit Tiny Patch 16 | Scratch | 13.34 | 21.61 | 27.27 | 31.65 | 34.93 |
| Mean (Sampled) | | | **39.76** | **47.46** | **51.47** | **53.98** | **55.83** |
| Mean (Scratch) | | | 17.43 | 27.97 | 33.91 | 37.82 | 40.43 |

### D.1.1 Performance after fixed number of optimization steps

Table 9: Model Performance across Optimization Steps. Accuracy (in %) of individual generated models after finetuning for for a fixed number of optimization steps with batch size 256 on ImageNet-1K when sampling from the large HF-backbone (Tab. 4) vs. training from scratch. The results show that the generated models outperform training from scratch in most cases after 50 steps but gains are architecture and scale dependent. On average over all architectures, generated models outperform randomly initialized models over all steps.

| Group | Architecture Name | Init | Steps 0 | 10 | 50 | 100 | 200 | 400 | 500 | 1000 |
|---|---|---|---|---|---|---|---|---|---|---|
| BEiT | Beitv2 Base 16 | Sampled | 0.10 | 0.11 | 0.42 | **0.59** | **1.22** | **2.22** | **2.91** | **5.00** |
| | Beitv2 Base 16 | Scratch | **0.11** | **0.23** | **0.54** | 0.58 | 1.07 | 1.84 | 1.91 | 2.92 |
| ConvNeXt | Convnext Base | Sampled | 0.10 | 0.10 | 0.14 | **1.44** | **6.26** | **17.75** | **42.96** | **52.55** |
| | Convnext Base | Scratch | 0.10 | **0.23** | **0.30** | 0.47 | 0.52 | 0.78 | 0.97 | 2.36 |
| | Convnext Small | Sampled | **0.10** | 0.10 | **0.56** | **1.50** | **4.80** | **22.58** | **25.85** | **51.02** |
| | Convnext Small | Scratch | 0.08 | **0.20** | 0.38 | 0.39 | 0.53 | 0.85 | 0.96 | 2.69 |
| | Convnext Tiny | Sampled | 0.14 | **13.31** | **60.89** | **64.31** | **66.58** | **67.17** | **68.37** | **68.84** |
| | Convnext Tiny | Scratch | 0.14 | 0.22 | 0.39 | 0.40 | 0.53 | 0.80 | 1.06 | 1.86 |
| DeiT | Deit3 Base Patch16 | Sampled | **0.14** | **0.15** | **0.35** | **0.57** | **1.28** | **2.15** | **3.04** | **5.18** |
| | Deit3 Base Patch16 | Scratch | 0.13 | 0.13 | 0.28 | 0.38 | 0.90 | 1.68 | 1.80 | 1.67 |
| | Deit3 Medium Patch 16 | Sampled | **0.15** | **0.25** | **0.49** | **1.15** | **2.17** | **4.58** | **5.60** | **9.06** |
| | Deit3 Medium Patch 16 | Scratch | 0.10 | 0.16 | 0.37 | 0.48 | 0.93 | 1.69 | 2.22 | 3.77 |
| DenseNet | Densenet121 | Sampled | **0.26** | **0.18** | **1.09** | **2.95** | **6.49** | **19.57** | **24.40** | **39.69** |
| | Densenet121 | Scratch | 0.14 | 0.11 | 0.33 | 0.70 | 1.11 | 1.99 | 3.10 | 5.38 |
| EfficientNet | Efficientnetv2 M | Sampled | **0.13** | 0.11 | **1.57** | **2.77** | **4.67** | **11.09** | **14.91** | **27.01** |
| | Efficientnetv2 M | Scratch | 0.10 | **0.16** | 0.19 | 0.32 | 0.43 | 0.87 | 1.36 | 3.64 |
| | Efficientnetv2 S | Sampled | **0.20** | **0.12** | **2.31** | **4.12** | **8.45** | **18.72** | **22.88** | **37.65** |
| | Efficientnetv2 S | Scratch | 0.08 | 0.11 | 0.15 | 0.42 | 0.62 | 1.62 | 2.11 | 5.09 |

Continued on next page

Table 9: Model Performance Across Optimization Steps (continued)

| Group | Architecture Name | Init | Steps | | | | | | | |
|-------|-------------------|------|-------|-----|-----|-----|-----|-----|-----|------|
| | | | 0 | 10 | 50 | 100 | 200 | 400 | 500 | 1000 |
| MobileNet | Mobilenetv2 100 | Sampled | 0.09 | **0.29** | **1.05** | **1.86** | **3.69** | **6.53** | **7.99** | **15.14** |
| | Mobilenetv2 100 | Scratch | **0.10** | 0.10 | 0.21 | 0.24 | 0.52 | 1.03 | 1.19 | 3.62 |
| | Mobilenetv3 Small 075 | Sampled | **0.15** | **0.26** | **0.73** | **1.46** | **2.16** | **4.84** | **6.18** | **11.69** |
| | Mobilenetv3 Small 075 | Scratch | 0.10 | 0.10 | 0.38 | 0.59 | 1.11 | 2.23 | 2.98 | 5.75 |
| | Mobilenetv3 Small 100 | Sampled | **0.11** | **0.20** | **0.68** | **1.30** | **2.00** | **5.17** | **5.88** | **10.92** |
| | Mobilenetv3 Small 100 | Scratch | 0.10 | 0.10 | 0.41 | 0.74 | 1.19 | 2.37 | 2.96 | 6.19 |
| ResNet | Resnet101 | Sampled | **1.02** | **1.12** | **21.13** | **49.74** | **59.44** | **64.75** | **67.24** | **68.92** |
| | Resnet101 | Scratch | 0.09 | 0.13 | 0.40 | 0.74 | 1.45 | 2.25 | 3.67 | 6.84 |
| | Resnet152 | Sampled | **0.14** | 0.12 | **0.63** | **2.28** | **7.83** | **32.12** | **40.39** | **55.44** |
| | Resnet152 | Scratch | 0.10 | **0.13** | 0.52 | 0.94 | 1.44 | 2.91 | 3.51 | 5.32 |
| | Resnet18 | Sampled | **0.15** | 0.14 | 0.31 | 0.83 | **2.62** | **7.36** | **13.70** | **37.00** |
| | Resnet18 | Scratch | 0.07 | **0.18** | **0.53** | **1.06** | 1.36 | 2.82 | 2.74 | 6.60 |
| | Resnet34 | Sampled | **0.16** | 0.11 | 0.39 | **0.81** | **3.24** | **13.99** | **23.73** | **44.88** |
| | Resnet34 | Scratch | 0.08 | **0.16** | **0.61** | 0.78 | 1.39 | 2.59 | 3.14 | 6.22 |
| | Resnet50 | Sampled | **7.09** | **29.17** | **60.30** | **64.16** | **67.87** | **68.83** | **69.48** | **70.64** |
| | Resnet50 | Scratch | 0.04 | 0.19 | 0.46 | 0.85 | 1.30 | 2.69 | 3.06 | 5.22 |
| Swin | Swin S3 Base | Sampled | **0.13** | 0.10 | **0.10** | **0.25** | **0.19** | **0.10** | **0.49** | **0.26** |
| | Swin S3 Base | Scratch | 0.10 | **0.18** | 0.08 | 0.16 | 0.09 | 0.10 | 0.14 | 0.10 |
| | Swin S3 Small | Sampled | **0.12** | 0.10 | **0.14** | **0.20** | **0.29** | **0.48** | **0.58** | **2.54** |
| | Swin S3 Small | Scratch | 0.06 | **0.16** | 0.13 | 0.13 | 0.15 | 0.10 | 0.10 | 0.10 |
| | Swin S3 Tiny | Sampled | **0.10** | 0.10 | 0.12 | **0.21** | **0.50** | **0.91** | **0.72** | **4.40** |
| | Swin S3 Tiny | Scratch | 0.09 | **0.19** | **0.23** | 0.18 | 0.10 | 0.15 | 0.27 | 0.10 |
| ViT | Tiny Vit 11M | Sampled | 0.09 | **0.17** | **0.81** | **1.06** | **1.26** | **2.81** | **3.38** | **9.69** |
| | Tiny Vit 11M | Scratch | **0.10** | 0.10 | 0.43 | 0.54 | 0.72 | 1.35 | 1.22 | 1.90 |
| | Tiny Vit 5M 224 | Sampled | **0.11** | **0.19** | **0.58** | 0.75 | **1.43** | **2.57** | **3.25** | **6.84** |
| | Tiny Vit 5M 224 | Scratch | 0.10 | 0.10 | 0.51 | **0.85** | 1.31 | 2.48 | 2.74 | 5.19 |
| | Vit Base Patch16 | Sampled | **0.19** | 0.13 | **0.38** | **0.98** | **2.97** | **5.24** | **5.33** | **17.35** |
| | Vit Base Patch16 | Scratch | 0.12 | **0.25** | 0.33 | 0.43 | 0.57 | 0.55 | 0.82 | 0.84 |
| | Vit Small Patch 16 | Sampled | **0.13** | **0.20** | **0.37** | **0.89** | **2.08** | **4.02** | **4.63** | **7.71** |
| | Vit Small Patch 16 | Scratch | 0.11 | 0.17 | 0.25 | 0.34 | 0.41 | 1.19 | 1.41 | 2.08 |
| | Vit Tiny Patch16 | Sampled | 0.11 | 0.10 | **0.45** | **1.22** | **2.98** | **5.87** | **7.39** | **12.91** |
| | Vit Tiny Patch16 | Scratch | **0.13** | **0.12** | 0.26 | 0.53 | 0.65 | 1.51 | 1.77 | 3.50 |
| Mean(Sampled) | | | **0.45** | **1.88** | **6.24** | **8.30** | **10.50** | **15.66** | **18.85** | **26.89** |
| Mean (Scratch) | | | 0.10 | 0.16 | 0.35 | 0.53 | 0.82 | 1.54 | 1.89 | 3.56 |

## D.2 DISCRIMINATIVE RESULTS

In Tab. 10 we evaluate whether the embeddings of the encoder trained on HF-data are still predictive of model properties as was shown in previous work (Schürholt et al., 2024). We observe a performance drop compared to previous work in all cases. Interestingly for CIFAR100 and TinyImageNet the $R^2$ remains competitive and about 5% lower compared to the baseline. This could possibly be attributed to the fact that the embeddings of the HF trained backbone have a higher variance than the single-zoo embeddings because to reconstruct models that vary in both architecture and training dataset accurately, more fine-grained information may be required. Improving the predictive performance in this setting may require going beyond a simple linear model better model non-linear relations between the embeddings and target properties. However, for CIFAR10 we observe a more significant drop

Table 10: Discriminative results on the ResNet-18 model-zoos. The results show explained variance ($R^2$) per target dataset. A linear probe fits the embeddings to the target properties of the respective trainset and the performance on the testset is reported. For the evaluation the ResNet-18 model zoos from the model zoo dataset are used (Schürholt et al., 2022c). 100 models are split into train/test/validation with proportions 70/15/15 using checkpoints from epochs 1, 3, 5, 10, 15, 20, and 25.

| Training Data | Test Accuracy | | | GGap | | | Epoch | | |
|---|---|---|---|---|---|---|---|---|---|
| | CIFAR10 | CIFAR100 | TIN | CIFAR10 | CIFAR100 | TIN | CIFAR10 | CIFAR100 | TIN |
| CIFAR10 | 91.69 | 95.60 | 94.99 | 75.93 | 91.94 | 88.81 | **99.67** | 99.34 | 99.11 |
| CIFAR100 | **92.70** | **96.22** | **95.73** | **77.56** | **92.21** | **88.90** | 99.65 | **99.54** | **99.30** |
| HF | 69.44 | 91.58 | 90.29 | 53.75 | 87.39 | 85.05 | 94.78 | 96.86 | 90.39 |

in performance that was also seen in the generative results which could be attributed to the fact that CIFAR10 trained models might be less prevalent on HF compared to CIFAR100 and TinyImageNet or that the models trained on CIFAR10 exhibit higher differences in terms of accuracy while still remaining close in weight space compared to the datasets with more classes.

## D.3 BASELINES

Our baseline results are divided into multiple parts. First we closely follow the experimental set-up of SANE (Schürholt et al., 2024) to validate our proposed loss normalization and compare with the results of the original SANE (see Sec. 4). To assess the performance of sampled models for our baseline when varying the dataset we train a single backbone per model zoo (CIFAR10 and CIFAR100) using the configuration detailed in Tab. 4 (SANE (MLN)). The other datasets that we sample models for are unseen during training to validate that our change to the loss normalization still allows generating models that are out of distribution. For the later experiments (App. D.3.2) we train the baseline with the same configuration as our Hugging Face backbone (Tab. 4, HF-Small) but still on a single model zoo (CIFAR10).

### D.3.1 DATASETS

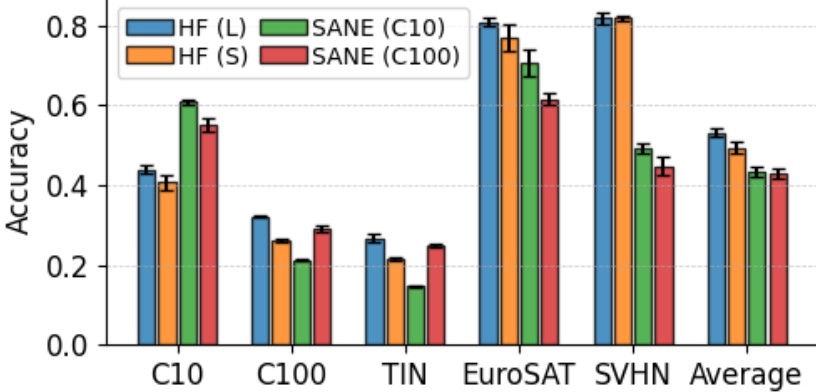

Figure 5: Performance of generated ResNet-18 models with varying backbones. Here we include the performance of both baselines separately whereas in the paper we show the max performance achieved over both baselines. Furthermore we show the performance of both the small and large HF backbone. The results indicate that training on HF models is feasible and outperforms the baselines with the exception of CIFAR10.

In the main paper (Fig. 2), we summarize the best performance achieved by each baseline model (trained individually on CIFAR10 and CIFAR100) and compare them to our large Hugging Face-trained backbone (HF-Large), using the configuration detailed in Tab. 4. In contrast, Fig. 5 provides

a more detailed view by showing the performance of both individual baselines and our smaller Hugging Face backbone (HF-Small). These results reveal that both HF-trained backbones remain competitive across the board. The large backbone outperforms the individual baselines on four out of five datasets, with CIFAR10 being the only exception. The model-zoo trained baselines perform well in-distribution and on datasets of comparable difficulty (i.e., similar number of classes), but their generalization to different datasets is more limited. In contrast, our HF-trained backbones are able to sample effectively across diverse datasets. Although HF-Large offers slight performance gains over HF-Small, it comes at the cost of increased training time. However, larger performance differences become apparent when sampling across architectures rather than datasets (see App. D.1).

### D.3.2 ARCHITECTURES

Table 11: Performance of individual generated models after finetuning for 1-5 epochs on ImageNet-1K when sampling from the SANE baseline trained on CIFAR10 (Tab. 4) compared to training from scratch. The baseline is competitive for architectures close to the model-zoo training set (e.g., ResNet-18,34,50 and a tiny ConvNext) but fails to scale to larger models as well as architectures less similar to those included in the training data such as transformers. For those architectures the performance is often even worse than training from scratch.

| Architecture Group | Architecture Name | Init | Epoch | | | | |
|---|---|---|---|---|---|---|---|
| | | | 1 | 2 | 3 | 4 | 5 |
| BEiT | Beitv2 Base Patch 16 | Sampled | 3.42 | 6.39 | 7.76 | 8.98 | 10.56 |
| | Beitv2 Base Patch 16 | Scratch | **6.87** | **9.94** | **9.48** | **10.98** | **12.24** |
| ConvNeXt | Convnext Base | Sampled | 0.10 | 0.10 | 0.10 | 0.10 | 0.10 |
| | Convnext Base | Scratch | **21.11** | **39.46** | **49.94** | **55.55** | **59.47** |
| | Convnext Small | Sampled | 0.17 | 0.10 | 0.10 | 0.10 | 0.10 |
| | Convnext Small | Scratch | **20.34** | **38.14** | **47.44** | **52.92** | **56.94** |
| | Convnext Tiny | Sampled | **25.30** | **41.68** | **50.28** | **55.03** | **57.84** |
| | Convnext Tiny | Scratch | 17.97 | 35.48 | 44.70 | 50.48 | 54.49 |
| DeiT | Deit3 Base Patch 16 | Sampled | 1.30 | 2.27 | 4.68 | 6.41 | 9.32 |
| | Deit3 Base Patch 16 | Scratch | **8.56** | **11.97** | **14.02** | **15.23** | **15.99** |
| | Deit3 Medium Patch 16 | Sampled | 1.28 | 2.98 | 4.83 | 6.95 | 8.83 |
| | Deit3 Medium Patch 16 | Scratch | **14.50** | **21.07** | **27.22** | **31.34** | **34.95** |
| DenseNet | Densenet121 | Sampled | 24.77 | 40.05 | 47.92 | 52.23 | 55.05 |
| | Densenet121 | Scratch | **28.49** | **41.57** | **49.32** | **53.75** | **55.79** |
| EfficientNet | Efficientnetv2 M | Sampled | 19.91 | 33.98 | 44.94 | 48.33 | 0.10 |
| | Efficientnetv2 M | Scratch | **25.88** | **41.20** | **47.77** | **54.04** | **57.59** |
| | Efficientnetv2 S | Sampled | 27.04 | **43.10** | 50.70 | 55.08 | 58.60 |
| | Efficientnetv2 S | Scratch | **28.84** | 42.86 | **51.55** | **55.83** | **59.18** |
| MobileNet | Mobilenetv2 100 | Sampled | **31.44** | **43.99** | **49.61** | **52.96** | **55.21** |
| | Mobilenetv2 100 | Scratch | 18.32 | 30.13 | 37.92 | 43.14 | 46.26 |
| | Mobilenetv3 Small 075 | Sampled | **24.66** | **35.77** | **41.29** | **44.35** | **47.49** |
| | Mobilenetv3 Small 075 | Scratch | 18.42 | 28.06 | 33.59 | 37.28 | 40.28 |
| | Mobilenetv3 Small 100 | Sampled | **27.62** | **38.59** | **43.70** | **47.15** | **49.30** |
| | Mobilenetv3 Small 100 | Scratch | 19.89 | 29.29 | 35.75 | 40.15 | 43.14 |
| ResNet | Resnet101 | Sampled | **34.14** | **50.52** | **59.29** | **63.28** | **63.45** |
| | Resnet101 | Scratch | 25.96 | 41.45 | 49.49 | 53.51 | 56.00 |
| | Resnet152 | Sampled | 23.95 | 40.07 | 46.98 | 52.36 | 56.03 |
| | Resnet152 | Scratch | **28.99** | **44.01** | **52.48** | **55.49** | **58.07** |

Continued on next page

Table 11: Performance of individual sampled models after finetuning for 1-5 epochs on ImageNet-1K when sampling from the SANE baseline trained on CIFAR10 (continued)

| Architecture Group | Architecture Name | Init | Epoch | | | | |
|---|---|---|---|---|---|---|---|
| | | | 1 | 2 | 3 | 4 | 5 |
| | Resnet18 | Sampled | **58.22** | **61.38** | **62.60** | **62.92** | **64.34** |
| | Resnet18 | Scratch | 18.30 | 32.25 | 38.77 | 43.78 | 47.86 |
| | Resnet34 | Sampled | **63.40** | **65.40** | **66.17** | **68.14** | **68.03** |
| | Resnet34 | Scratch | 23.24 | 32.18 | 41.65 | 47.41 | 49.10 |
| | Resnet50 | Sampled | **44.82** | **58.99** | **63.16** | **65.43** | **67.74** |
| | Resnet50 | Scratch | 23.92 | 40.10 | 46.63 | 51.45 | 54.00 |
| Swin | Swin S3 Base 224 | Sampled | 0.10 | 0.10 | 0.10 | 0.10 | 0.10 |
| | Swin S3 Base 224 | Scratch | 0.10 | 0.10 | 0.10 | 0.10 | 0.10 |
| | Swin S3 Small 224 | Sampled | 0.10 | 0.10 | 0.10 | 0.10 | 0.10 |
| | Swin S3 Small 224 | Scratch | 0.10 | 0.10 | 0.10 | 0.10 | 0.10 |
| | Swin S3 Tiny 224 | Sampled | 0.10 | 0.10 | 0.10 | 0.10 | 0.10 |
| | Swin S3 Tiny 224 | Scratch | 0.10 | 0.10 | 0.10 | 0.10 | 0.10 |
| ViT | Tiny Vit 11M 224 | Sampled | 0.10 | 0.10 | 0.10 | 0.10 | 0.10 |
| | Tiny Vit 11M 224 | Scratch | **24.42** | **42.43** | **49.50** | **54.88** | **57.24** |
| | Tiny Vit 5M 224 | Sampled | 0.79 | 4.04 | 11.82 | 18.16 | 22.90 |
| | Tiny Vit 5M 224 | Scratch | **29.08** | **43.20** | **49.88** | **53.96** | **56.12** |
| | Vit Base Patch 16 224 | Sampled | 3.34 | 5.82 | 8.17 | 10.17 | 11.71 |
| | Vit Base Patch 16 | Scratch | **7.67** | **13.49** | **18.14** | **22.04** | **26.68** |
| | Vit Small Patch 16 | Sampled | 3.73 | 8.14 | 12.31 | 16.28 | 20.77 |
| | Vit Small Patch 16 | Scratch | **11.20** | **18.94** | **25.03** | **30.27** | **34.23** |
| | Vit Tiny Patch16 224 | Sampled | **14.17** | **24.33** | **31.57** | **36.49** | **40.04** |
| | Vit Tiny Patch 16 | Scratch | 13.34 | 21.61 | 27.27 | 31.65 | 34.93 |
| Mean Accuracy (Sampled) | | | 17.36 | 24.32 | 28.33 | 30.85 | 30.72 |
| Mean Accuracy (Scratch) | | | **17.43** | **27.97** | **33.91** | **37.82** | **40.43** |

## D.4 ABLATIONS

### D.4.1 MASKED LOSS NORMALIZATION (MLN)

To further investigate whether MLN can be used as a suitable replacement for layer wise loss normalization, we evaluate the distribution of reconstructed weights vs original weights when not normalizing the loss at all, normalizing the loss per-token (including masked values) and normalizing on signal values only. After training, we use models from the test split to be reconstructed by the backbone (which corresponds to a simple forward pass through the encoder-decoder). Following previous work, we use the match of weight distribution as a proxy for how well-reconstructed models mirror the original models (Schürholt et al., 2022a). Results are shown in Fig. 6.

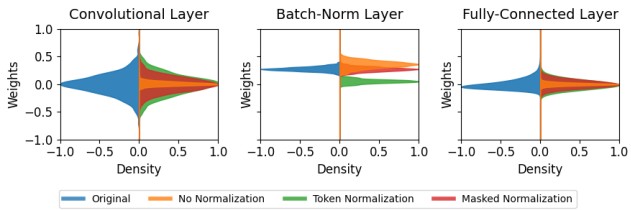

Figure 6: Comparison of weight distributions of a selection of ResNet layers between original weights (blue/left) vs reconstructed weights (right). We compare reconstruction without normalization (orange), with full token normalization (green) and with masked loss normalization (red). Without normalization the weights of layers with narrow distributions are squashed towards the mean. Normalizing per-token fixes that issue. Ignoring the mask introduces a strong bias, particularly for batch-norm layers. Reconstructions with MLN match the original the closest.

Masked loss normalization achieves a
more accurate alignment between the reconstructed distribution and the original weight distribution across model parameters compared to full-token normalization, see Fig. 6, particularly of batch-norm layer weights. By focusing on signal values only, the masked normalization more effectively maintains the original weight distributions, reducing reconstruction error and providing a stable signal even in high-parameter regimes.

### D.4.2 TOKENIZATION

In this section we compare the two different tokenization variants introduced in Sec. 3.3. We evaluate whether there are any significant differences in terms of pretraining and downstream performance when training on HF models. To that end, we compare the explained variance $R^2 = 1 - \frac{\sum_i \|T_i - \hat{T}_i\|^2}{\sum_i \|T_i - \bar{T}\|^2}$ of the reconstructed tokens to validate if our backbone converges when training on the HF dataset. As discussed previously, sparse tokenization can add substantial amounts of padding per token when tokenizing different architectures. More specifically, after tokenization our HF model dataset includes approximately 600M tokens for the dense variation and 730M tokens using sparse tokenization (with tokensize 288). The largest model included in the HF trainset contains ~1.3B weights and is split into 5M individual tokens using sparse and 4.5M tokens using dense tokenization. Since the models have vastly different sizes, we use the number of tokens in the dataset as measure of size.

**Dense tokenization matches performance while being significantly more efficient** In our HF dataset, sparse tokenization introduces 20% padding per token, whereas dense tokenization adds only 0.01% padding on average. As a result, when processed by the weight-space backbone, dense tokens lead to a higher compression ratio compared to sparse tokens. To ensure a fair comparison, we adjust the token size of the dense dataset accordingly. The results in Fig. 7 show that both tokenization strategies yield similar reconstruction quality during backbone training. In addition, dense tokenization reduces disk usage by 20% relative to sparse tokenization, leading to faster training and offering a better balance between efficiency and performance. We further evaluate and discuss downstream performance of sampled models across five datasets when varying tokenization in App. D.1.

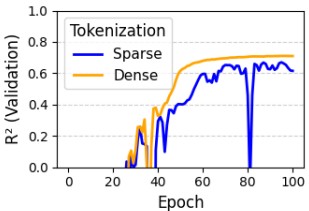

Figure 7: $R^2$-score of reconstructed tokens on a hold-out validation set of models to assess reconstruction quality. The results indicate that dense tokenization works and shows ~~~~ other convergence.

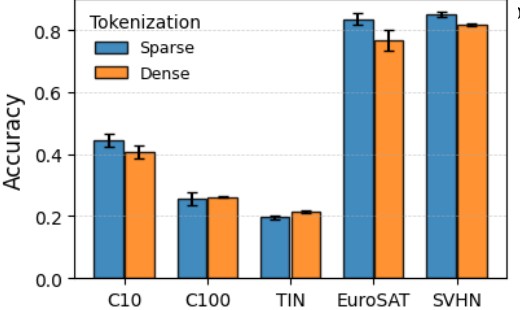

Figure 8: Performance of sampled models of our HF-Small backbone configuration when using sparse or dense tokenization.

In Fig. 8 we show the accuracy of sampled models on the target
dataset without any finetuning. To train the backbones we use the
configuration as outlined in Tab. 4. The results show higher performance when using sparse tokenization for the datasets with ten classes whereas dense tokenization shows slightly higher performance for the datasets with 100 and 200 classes. However, as mentioned in the main paper, dense tokenization is more efficient during training and we believe that the overhead introduced with sparse tokenization in terms of storage and compute is not preferable to dense tokenization overall, in particular for our training set-up with diverse architectures in the training dataset.

### D.4.3 POSITIONAL ENCODINGS

For evaluating the performance impact of using sinusoidal positional encodings instead of learned embeddings we train the weight space backbone in the HF-small configuration (Tab. 4) on model zoos as we cannot train of the HF dataset with learned embeddings. The results show we achieve similar performance compared to the baseline setting while only training a single backbone instead of one per dataset/architecture pair. We do observe however that in this setting we need to train for more epochs compared to using learned embeddings.

Table 12: Performance of sampled ResNet-18 models. We compare the baseline setting with learned positional embeddings to our HF configuration trained on model zoos.

| Configuration | Pos Embed | CIFAR10 | CIFAR100 | TIN |
|---|---|---|---|---|
| SANE (MLN) | Learned | **68.6±1.2** | 20.4±1.3 | 11.7±0.5 |
| HF (S) | Sinusoidal | 66.8±0.7 | **27.93±0.7** | **16.05±0.2** |

### D.4.4 TRAINING DATASET COMPOSITION

Table 13: Performance of sampled ResNet-18 models when training only on a fraction of the data. The results indicate that after a certain number of training tokens the performance saturates and does not increase significantly anymore given a constant backbone size. On the other hand, performance drops to random guessing (with the exception of EuroSAT and SVHN) when training on less than ~45M tokens showing that the number of samples included are more important than trying to restrict the models in the training data to be more similar in terms of architecture, even if we use the same architecture during training and sampling. Furthermore, training on HF data requires more samples than when training on a homogeneous model zoo, as the HF data is more noisy and possibly also contains models that are not converged or only trained for a few epochs. Nevertheless, when including enough samples training on HF is competitive and even outperforms training on homogenous model zoos in most cases.

| Data Fraction | Num Tokens (~) | CIFAR10 | CIFAR100 | TIN | EuroSAT | SVHN |
|---|---|---|---|---|---|---|
| 1 | 590M | **31.38±3.91** | 32.24±1.34 | 20.14±0.88 | **78.66±0.81** | **83.96±1.23** |
| 0.64 | 388M | 30.02±4.50 | **33.27±1.49** | **23.10±1.33** | 78.01±1.40 | 82.35±1.03 |
| 0.32 | 189M | 21.90±1.59 | 27.98±1.70 | 17.78±1.10 | 79.57±3.10 | 82.84±0.46 |
| 0.16 | 95M | 16.52±0.96 | 26.28±0.83 | 16.34±0.74 | 79.50±1.95 | 83.18±0.62 |
| 0.08 | 47M | 11.39±0.68 | 5.45±0.56 | 1.78±0.18 | 74.23±1.79 | 78.19±1.34 |
| 0.04 | 24M | 10.63±0.27 | 1.14±0.07 | 0.50±0.04 | 52.28±4.42 | 45.04±2.58 |
| 0.02 | 12M | 11.78±0.69 | 1.15±0.16 | 0.55±0.06 | 14.58±2.15 | 19.54±0.08 |
| 0.01 | 6M | 10.50±0.27 | 1.14±0.13 | 0.51±0.03 | 11.12±0.41 | 19.59±0.00 |

In Tab. 13 we train only on the specified fraction of the dataset using a logarithmic scale from 0.01 to 0.64. For reference we also include the previous result acquired by training on the full vision transformer HF datset. The results indicate that increasing training sample count is beneficial up to a certain degree. The backbone trained on 64% of the data performs similarly or even better in some cases compared to training on the full dataset, which also serves as motivation for training a larger backbone given there is enough data available. Performance drops significantly when further restricting the number of samples and remains around random guessing when training on 24M or fewer tokens with the exception of EuroSAT and SVHN, similar to the performance of the HF ResNet trained backbone that contains 35M tokens. Therefore it is beneficial to include more samples and more diverse architectures over restricting the dataset to a single architecture class if there are not enough samples available.

