# OpenReview forum: "Learning Neural Representations From Publicly Available Model Hubs"
_ICLR.cc/2026/Conference — Submitted to ICLR 2026_

### Official Review · Reviewer_W9pP · 2025-10-29

**Soundness:** 1
**Presentation:** 3
**Contribution:** 2
**Rating:** 2
**Confidence:** 5

**Summary:**

The paper proposes a new approach for training encoder-decoder based weight-space algorithms. Instead of training on lab curated models the authors propose to train on publily available models from huggingface with heterogeneous architectures and tasks. They make adjustments to the SANE architecture for learning on large scale and mixed architectures by adding a loss normalization, modified tokenization and sinusoidal positional encoding. The experimental section tests the new approach against SANE on several weight generation tasks.

**Strengths:**

- The paper targets an important and emerging task of learning from a collection of model weights to generalize to newly created models.

- The idea of learning from heterogeneous model collections at once is interesting.

- The paper is clearly written and easy to follow.

**Weaknesses:**

- **Experimental design and results.**
  To my understanding of the experimental section, some of the empirical findings are odd. Specifically, I am having trouble to understand how the model samples reasonable classifiers for tasks without knowing for which downstream task the sampled model is aimed for. To give an example, in Tab. 2, a metnetworkd is trained on the HF collection have seen models from variety of tasks (ImageNet, CIFAR10, CIFAR100). But, when sampling a new network, the metanetwork does not know if the sampled calssifier would be used for ImageNet, CIFAR10 or CIFAR100. Hence, since there is not additional fine-tuning, how can the metanetwork sample reasonable calssifiers for all tasks if the sampling process stays the same? This also affects the other training collections in Tab. 2 which are also composed of models trained for differnt tasks.

  Moreover, this issue is even more prominent in the experiments of Fig. 2, where there are more downstream tasks including ones that have not been seen by the training models at all. For example, consider the SANE baseline which was trained only on the CIFAR10 and CIFAR100 model zoos. No model in its training set have been trained on the EuroSAT dataset. So, how can it generate accurate classifiers for EuroSAT (without further fine-tuning) if it doesnt even know the EuroSAT classes order (which can be chosen arbitrarily)?

  Could the authors please elaborate on these issues? If I misunderstood some parts of the experimental design please let me know so I can reconsider.

- **Generalization or memorization.** A recent work by Zeng et al. [1] demonstrated that weight generation experiments should be carefully examined, as metanetworks are prone to memorization instead of generalization. Therefore, to estimate the merit of the proposed approach for weight generation, its generated networks should be compared to the best pre-trained network in its training collection. For example, in Tab. 1 an additional baseline should be the best networks for CIFAR10, CIFAR100 and TIN from the training datasets. I believe this affects most of the generative results presented in the main menuscript including Tab. 1 as well as Fig. 3 where these baseline networks might provide better initialization for further fine-tuning.

- **Differences from SANE.** The modifications of the proposed approach over SANE seem relatively small. Tab. 1 also shows that in some cases these modifications can decrease performance when applied to SANE.


[1] Zeng, Boya, et al. "Generative Modeling of Weights: Generalization or Memorization?." arXiv preprint arXiv:2506.07998 (2025).

**Questions:**

- In Fig. 4 did the authors try to initialize the GPT model weights from randomly selected weight matrices from the training data?

---

> ### Author Response · Authors · 2025-11-21
>
> We would like to thank reviewer W9pP for their review, and will try to address the questions and concerns they raised.
>
> > Experimental design and results
>
> The reviewer is correct that the backbone itself does not receive any downstream task identifier. As in prior work [2], the sampling procedure is conditioned on prompt models used as  anchors, not on a task label. At generation time, we select a small number of anchor models from the target architecture and dataset, tokenize them, project them to the latent space, and fit a KDE over their latent embeddings. The KDE defines a local latent distribution encoding dataset information therefore samples from this latent distribution tend to decode into models that inherit the coarse structure and performance characteristics of the anchors.
>
> In other words, the weight-space backbone does not produce a universal classifier that works for all datasets. Instead, the downstream task is implicitly specified by the choice of anchors used to build the KDE. The HF-trained backbone provides a shared latent space and a decoder to translate tokens from this latent into model weights; the anchor-based KDE provides the task-appropriate conditioning. We use this evaluation to test whether the backbone is able to learn a meaningful representation when trained in such a heterogenous, noisy setting. We will clarify this more explicitly in the paper: generation is not unconditional, nor does our method claim to infer unseen class semantics.
>
> > Generalization or memorization
>
> We agree that memorization vs generalization in these settings can be an issue and that [1] raises an important discussion for many of the existing weight-generative methods. Our setting differs from theirs in a few key ways:
>
> Zeng et al. mainly study generative models trained on checkpoints from the same or closely related architectures (often many checkpoints from the same training run and with very small models as training data). In contrast, we train a single autoencoder across 2000 heterogeneous HF models spanning multiple architecture families (ResNet, ConvNeXt, ViT, Swin, etc.), many of which have no architectural match with the target model at test time. Furthermore, for the experiments where we sample ResNet-18 models for different datasets, none of the anchors are contained in the training set since these are taken from the model zoo dataset [3] and are not hosted on HuggingFace.
>
> > Differences from SANE
>
> While the backbone is based on SANE, our method involves non-trivial modifications required for training on uncurated HF data, a much harder setting compared to training on curated model zoos. We do not claim fundamental architectural novelty per se, but rather methodological robustness under real-world constraints. To the best of our knowledge, this is the first work demonstrating that training of a shared latent representation from heterogeneous repositories as Hugging Face is possible.
>
> ---
>
> [1] Zeng et al., (2025) "Generative Modeling of Weights: Generalization or Memorization?." arXiv preprint arXiv:2506.07998.
>
> [2] Schürholt et al., (2024). "Towards Scalable and Versatile Weight Space Learning.", ICML 2024.
>
> [3] Schürholt et al., (2022). Model zoos: A dataset of diverse populations of neural network models. NeurIPS 2022

---

### Official Review · Reviewer_nTRj · 2025-10-30

**Soundness:** 2
**Presentation:** 3
**Contribution:** 2
**Rating:** 4
**Confidence:** 4

**Summary:**

This paper proposes to use HuggingFace (HF) as a source of trained neural networks for weight space learning (WSL). Previous papers had previously designed model zoos for WSL. To do this, the paper proposes a new method to train a shared weight space representation from arbitrary models downloaded from HF, using a novel method that is capable of learning from models with arbitrary architectures.

**Strengths:**

Strengths of this paper include:

**(S1)**: This paper proposes to leverage heterogeneous datasets of pre-trained models instead of curated model zoos, and proposes a new dataset curated from HuggingFace

**(S2)**:  This paper also proposes a new method to adapt existing works (i.e. SANE) to the challenge of heterogeneous model zoos using a new tokenization method and new normalization scheme

**(S3)**: The paper is well-written and clear, with many of the details of the method provided

**Weaknesses:**

**Major Weaknesses**

**(W1) Clarity**: The paper could be more clear in the following areas:
1. **Single weight space representation**: does this mean that there is a shared representation space for all models, regardless of architecture/training dataset, but each model has a unique representation in this space? Or does this mean that multiple unique models share the same representation?
2. **Evaluation protocol**: I think that the paper could be made more self-contained by putting more details on both the experimental procedure and evaluation and how models were generated in the main text of the paper.

**(W2) Limited diversity of experiments**: The experiments are limited to examining only the accuracy of generated models.

**(W3) Claim of generalization to new modalities/OOD tasks**: I understand that it may be difficult to validate how all the different HF models were trained if the details weren’t provided, but I don’t think it’s fair to make this claim without checking to see if any of the models were trained with language embeddings or only training with models whose training protocol is clearly stated, such as curated model zoos. Currently, there is no baseline showing that a lab-produced model zoo does not also have this behavior. Also, there are no comparisons to other weight-initialization methods.

**(W4) KDE bandwidth**: The influence of the KDE bandwidth parameter for model generation (Appendix B.2.2) is not examined, and the values of this parameter are not reported in the paper.

**Minor Weaknesses**
Typos:
1. Line 220: “with a focus processing” -> “with a focus on processing”
2. Table 2: At the intersection of the “all” row and the “Num models” columns, it seems that it should be 2666 and not 2000?

**Questions:**

**(Q1)**: Random Fourier Features (RFF) positional encodings (PE) do not require additional trainable parameters and in many cases are better than sinusoidal positional encodings. Would they work better in this case as well?

**(Q2)**: Using MLN instead of LWLN and sinusoidal PE instead of learnable PE leads to lower performance on CIFAR-10, is there a reason why this is?

**(Q3)**: Is “single weight space representation” synonymous with having a shared representation space among all models, regardless of architecture/training dataset?

**(Q4)**: For each dataset, how does the accuracy of generated models compare to the accuracy of the HuggingFace models they were trained on?

**(Q5)**: What are the performance for curated vision model zoos on the GPT-2 task (e.g. Section 4.4.1, Figure 4)

**(Q6)**: The performance of generated models is much lower than what you would expect from just training a model on, say, CIFAR-10. What is the reason for this? What are the failure modes of these models?

**(Q7)**: Is there any structure to the latent representation space of the proposed model? Are the latent spaces of this model vs SANE meaningfully different? Do model with similar architectures or similar training datasets get mapped to similar regions of latent space?

---

> ### Author Response · Authors · 2025-11-21
>
> We would like to thank reviewer nTRj for their review, and will try to address the questions and concerns they raised.
>
> > (Q1): Random Fourier Features (RFF) positional encodings (PE) do not require additional trainable parameters and in many cases are better than sinusoidal positional encodings. Would they work better in this case as well?
>
> Thank you for the interesting suggestion. We did not experiment with Random Fourier Feature positional encodings and therefore cannot say whether they would work better in this case as well. They could be a promising alternative, and are indeed interesting because they avoid learned position tables. In our setting, the primary requirement was a simple, well established, parameter-free encoding that scales to very long sequences and arbitrary model sizes, which sinusoidal embeddings satisfy.
>
> > (Q2): Using MLN instead of LWLN and sinusoidal PE instead of learnable PE leads to lower performance on CIFAR-10, is there a reason why this is?
>
> A potential reason for the lower performance on CIFAR-10 when using MLN compared to SANE is that we apply the same training configuration across all model zoos, regardless of dataset complexity. While this uniform setup simplifies training, it could be further tuned per model zoo to maximize performance. Furthermore, we view the proposed masked per-token loss normalization (MLN) as an approximation of previous per-layer normalization schemes that enables training directly on heterogeneous model collections without explicit per-layer weight normalization during preprocessing. This makes it possible to train on arbitrary models from sources like Hugging Face, where models do not necessarily have the same layers and for which per-layer loss normalization is therefore not applicable.
>
> > (Q3): Is “single weight space representation” synonymous with having a shared representation space among all models, regardless of architecture/training dataset?
>
> Yes, by “single weight space representation” we mean that we train one shared backbone that can embed and reconstruct any model in the heterogeneous HF collection, regardless of its architecture, training dataset or task and without architecture specific preprocessing and weight normalization. This allows us to use the same backbone for all experiments instead of training individual weight-space backbones per architecture/dataset family as done in prior work [1, 2, 3].
>
> > (Q4): For each dataset, how does the accuracy of generated models compare to the accuracy of the HuggingFace models they were trained on?
>
> That’s a good question, we know that for a large fraction of Hugging Face there is no documentation, making it very difficult to compare. For the other part of the Hugging Face model collection, we believe that indeed individual models could be better in performance than the ones we sampled. There is discussion in the weight space learning community that retrieval of suitable models from Hugging Face is a valid alternative to weight generation of new models. We share this opinion and at the same time think that sampling models would allow us to interpolate in latent weight space to sample models for non-standard datasets, i.e, models trained on datasets that are not part of Hugging Face. For this scope, we believe that this work provides a first step in this direction.
>
> > (Q5): What are the performance for curated vision model zoos on the GPT-2 task (e.g. Section 4.4.1, Figure 4)
>
> Thank you for the suggestion. Based on the results in Sec. 4.4 where we have seen that the SANE baseline is only able to generate weights for architectures that are close to the models from the training zoo (e.g. a small ConvNext model or ResNet models) and for more different and larger architectures (e.g. ViT-B) leads to worse performance than random initialization; we expected that the more challenging GPT-2 task will also underperform. We have tested this and the results are indeed also below the performance of randomly initialized models when using the SANE baseline to generate the GPT-2 weights. However, we do agree that verifying that the HF backbone generalizes in this setting is difficult and was not the intent. We will rename the section and revise the text accordingly.
>
> ---
>
> [1] Wang et al., (2025). "Recurrent diffusion for large-scale parameter generation." arXiv preprint arXiv:2501.11587.
>
> [2] Schürholt et al., (2024). "Towards Scalable and Versatile Weight Space Learning.", ICML 2024.
>
> [3] Bedionita et al., (2025). "DIFFUSION-BASED NEURAL NETWORK WEIGHTS GENERATION." ICLR 2025.

---

> > ### Author Response · Authors · 2025-11-21
> >
> > > (Q6): The performance of generated models is much lower than what you would expect from just training a model on, say, CIFAR-10. What is the reason for this? What are the failure modes of these models?
> >
> > The lower zero-shot performance is expected for two main reasons. First, the backbone is trained only on raw weights and receives no signal from downstream images, labels, or task-specific loss functions. This places it at a fundamental disadvantage compared to standard training of a single model, where gradients from the target data can be used directly. Second, the HF training corpus is highly heterogeneous: architectures, datasets, and training protocols vary widely, and the representation must capture structures that are shared across this diversity rather than those optimized for any single dataset such as CIFAR-10.
> >
> > > (Q7): Is there any structure to the latent representation space of the proposed model? Are the latent spaces of this model vs SANE meaningfully different? Do model with similar architectures or similar training datasets get mapped to similar regions of latent space?
> >
> > Thank you for the interesting suggestion. Analyzing latent-space structure is challenging and not a primary objective of the paper. Our focus here is functional, in particular whether a single backbone can encode and generate across heterogeneous sources, not on enforcing or analyzing latent topology. We do agree that this could be a very interesting study in future work.

---

### Official Review · Reviewer_cXqv · 2025-10-31

**Soundness:** 3
**Presentation:** 3
**Contribution:** 3
**Rating:** 6
**Confidence:** 5

**Summary:**

The authors propose to learn weight-space representations directly from arbitrary models scraped from large public hubs (Hugging Face) rather than from curated, homogeneous “model zoos.” Concretely, the authors adapt a SANE-style encoder–decoder transformer to be architecture- and dataset-agnostic by introducing

(i) Masked Loss Normalization (MLN) that normalizes reconstruction loss per token at runtime while respecting token masks,

(ii) a dense tokenization scheme that reduces padding across heterogeneous layers/architectures, and

(iii) sinusoidal positional encodings to avoid massive learned position tables that would otherwise explode with long weight sequences.

Trained on ~2,000 HF models (≈171B parameters total), a single backbone is used for multiple downstream tasks, mainly generative initialization of diverse architectures/datasets. It often outperforms backbones trained on lab model-zoos, and shows OOD modality generalization by initializing GPT-2 better than training from scratch after short finetuning. The authors provide ablations on MLN, tokenization, positional encodings, and training-set composition, plus comparisons against SANE baselines.

**Strengths:**

- Heterogeneous-hub training at scale. Training a single model on ~2k HF models (≈171B params) while remaining architecture and dataset agnostic is a practical contribution that lowers the barrier to WSL research.

- Method tweaks that matter. MLN effectively replaces per-layer normalization and is demonstrated to stabilize learning and improve generation on harder datasets; dense tokenization and sinusoidal PEs cut padding/params and enable scaling to larger, varied architectures.

- Broad generative evaluation. Results span multiple datasets (CIFAR-10/100, Tiny-ImageNet, SVHN, EuroSAT) and many architectures (ResNet/ConvNeXt/EfficientNet/ViT/Swin/BeiT/GPT-2), generally showing faster/better finetuning vs. scratch, and gains over SANE zoo-trained baselines in most cases.

- Ablations and implementation detail. Clear hyperparameters, runtime footprint, and ablations (MLN vs. LWLN, tokenization, PEs, dataset fraction and composition).

**Weaknesses:**

- Zero-shot quality seems weak; benefits hinge on finetuning. For many larger models, initial accuracy is near random, with benefits materializing only after a few epochs. This limits plug-and-play use compared to stronger weight generators or data-driven pretraining. I would recommend the authors to report more stringent “few-step” regimes and to compare sample-efficiency curves, which can help strengthen the paper further.

- The reported results hider limited discriminative validation. Encoder embeddings trained on HF are worse for property prediction than zoo-trained baselines. The authors hypothesize higher variance and non-linearity but do not probe fixes (nonlinear heads, better pooling, contrastive variants). This somewhat weakens the “single backbone” versatility claim.

- Data curation and licensing clarity. The download protocol excludes remote-code models and malformed checkpoints, but licensing/provenance and potential duplication/fine-tune lineage (e.g., many HF models fine-tune of the same base) aren’t deeply analyzed. These could bias and hurt the learned representation while raising reuse concerns. (The Discussion notes possible selection bias but does not quantify it.)

- Comparative baselines. The main comparison is with respect to SANE (under the authors modifications). It would strengthen the case to include diffusion-based and/or graph/functional backbones (even on smaller subsets), or to report hybrid (e.g., diffusion in latent-Z) to show trade-offs. The current evidence supports feasibility but not superiority across the method classes.

- Metrics/uncertainty. Several plots lack error bars. The tables sometimes report ± but figures do not, making it hard to assess variance across seeds and models. More consistent reporting would help.

**Questions:**

- MLN vs. alternatives. Did the authors try token-wise standardization in the forward path (like a learnable norm layer) instead of normalizing only in the loss? Any effect on stability/quality? Would be nice to see/include a short ablation.

- Are there any dense tokenization trade-off here?. Dense tokens mix channels across layers. Do the authors observe any loss of locality that harms small models? Would be nice that the authors report sparsity vs. quality vs. memory curves contrasting dense vs. sparse across 2–3 architectures?

- Zero-shot vs. few-shot. How do results change under N optimizer steps (N∈{0,10,50,100}) with fixed batch size across all architectures? A unified “steps-to-X%” metric would clarify sample-efficiency gains over scratch.

- Encoder utility. In the reported results, the HF-trained encoder underperforms for property prediction. Have the authors tried non-linear heads, token-attention pooling, or contrastive objectives tuned for discriminative tasks? Would be nice that the authors elaborate/explain on that more (ideally would be nice that they provide a small table).

- Cross-modality support. For GPT-2, could the authors report exact init metrics (loss/perplexity at step 0) and early convergence deltas vs. scratch across 3 seeds? Also, could the authors explan more about do CLIP-like weights in the HF set confound the “vision-only” claim?

- Dataset hygiene. Can you quantify duplication (identical or near-identical checkpoints), license coverage, and model lineage (fine-tunes of the same base) in your 2k-model set? A brief audit would help assess bias and legal reusability.

- Backbone scale. Seems like HF-Large (\~900M) improves results over HF-Small (\~456M). Could the authors sat more about where are the diminishing returns? Any evidence that position encodings or window size become the bottleneck next?

**Details Of Ethics Concerns:**

/

---

> ### Author Response · Authors · 2025-11-21
>
> We would like to thank reviewer cXQV for their review, and will try to address the questions and concerns they raised.
>
> > MLN vs. alternatives. Did the authors try token-wise standardization in the forward path (like a learnable norm layer) instead of normalizing only in the loss? Any effect on stability/quality? Would be nice to see/include a short ablation.
>
> Thank you for the interesting suggestion. We did not experiment with learnable token-wise normalization layers in the forward pass. Our focus was on MLN because it is architecture-agnostic, leaves raw weight statistics unchanged during encoding/decoding, and enables heterogenous training at scale. A controlled comparison of forward-path standardization will probably require additional training runs beyond what could be done during the rebuttal period.
>
> > Are there any dense tokenization trade-off here?. Dense tokens mix channels across layers. Do the authors observe any loss of locality that harms small models? Would be nice that the authors report sparsity vs. quality vs. memory curves contrasting dense vs. sparse across 2–3 architectures?
>
> There is indeed a tradeoff to using the dense tokenization. In a controlled model zoo setting, where the token size can be tailored to each architecture, dense tokenization does perform worse compared to the sparse tokenization scheme and requires longer training. In the Hugging Face setting, however, the situation is different because architectures vary widely in shape. Sparse tokenization introduces substantial padding, around 20 percent on average in our collection, which increases memory usage and sequence length. This results in roughly 20 percent longer training. In this heterogeneous scenario, we observe only minimal differences between dense and sparse tokenization in downstream accuracy of generated models and in terms of explained variance during pretraining (see also App. D.4.2).
>
> Below we show the padding percentage when using the same tokensize for a small sample of different architectures to illustrate this:
>
> | Model              | Dense Tokenization | Sparse Tokenization |
> |--------------------|--------------------|----------------------|
> | ConvNeXt-S         | 0.04%              | 27.88%               |
> | ResNet-34          | 0.02%              | 10.63%               |
> | ViT-B/16           | 0.01%              | 13.55%               |
> | MobileNet-V3-0.75  | 0.58%              | 61.43%               |
>
> > Encoder utility. In the reported results, the HF-trained encoder underperforms for property prediction. Have the authors tried non-linear heads, token-attention pooling, or contrastive objectives tuned for discriminative tasks? Would be nice that the authors elaborate/explain on that more (ideally would be nice that they provide a small table).
>
> We agree that our backbone underperforms compared to model-zoo-trained SANE models, since the focus in this work is on assessing whether a single backbone trained on heterogeneous HF models can achieve comparable performance, we want to keep downstream tasks configurations as close as possible to previous work. For this reason, we intentionally used the same simple linear probe setup as prior work. As shown in Table 9, the HF-trained encoder remains competitive on CIFAR100 and TinyImageNet, with a modest drop relative to zoo-trained baselines, but performs worse on CIFAR10, which is consistent with the generative results and possibly reflects the lower prevalence and higher variability of CIFAR10 models in HF.
>
> We did not experiment with non-linear heads, or token-attention pooling, although these are promising avenues that could possibly recover part of the gap. Besides the goal of staying close to the original SANE setup for comparison’s sake, it is common practice to use linear heads to evaluate representations for discriminative downstream tasks [1]. Furthermore, we did optimize our hyper-parameters for generative downstream tasks as we focus our evaluation on weight generation. With different settings (e.g. more weight on the contrastive loss as the reviewer suggested) this could likely be further optimized for discriminative tasks.
>
> ---
>
> [1] Kornblith et al., "Do better imagenet models transfer better?." CVPR2019

---

> > ### Author Response · Authors · 2025-11-21
> >
> > > could the authors explain more about do CLIP-like weights in the HF set confound the “vision-only” claim?
> >
> > To construct our model collection we query Hugginface for models that are tagged as computer vision models (specifically, image classification and segmentation, object detection, depth estimation, see Sec. 2) only and therefore do not include language models or generative image models in the query. Given that the goal of our paper is to learn from arbitrary models, we do not employ further filtering of checkpoints beyond basic feasibility checks. Since HF models are inconsistently documented, it is difficult to guarantee that none of the downloaded vision models contain components trained jointly with text embeddings (for example, CLIP-style backbones as outlined in the paper). We were able to identify 40 out of 2000 models that seem to be trained with clip, but cannot guarantee that there are not more (we want to be very transparent here). The idea behind the GPT-2 experiment is to show that the backbone can handle a different modality during generation, irrespective of the models it was trained on. In our case, it can provide a useful initialization, even when the backbone is trained primarily on vision models. Assessing the level of generalization in this setting is difficult. To avoid confusion and overclaiming we will rename the section to “Generating Weights for a Different Modality” in the upcoming revision and revise the text.
> >
> > > Dataset hygiene. Can you quantify duplication (identical or near-identical checkpoints), license coverage, and model lineage (fine-tunes of the same base) in your 2k-model set?
> >
> > In our collection 60% of models are covered under an apache or mit license and about one third has no specific license attached. We only use models that are available publicly in non gated repos via HuggingFace API. Quantifying duplicates and model lineage is difficult in this setting. As we show in section two, the dataset covers a wide range of architectures and in general for these discriminative CV models we believe that finetuned models of the same base are less common compared to generative image models or large language models.
> >
> > > Backbone scale. Seems like HF-Large (\~900M) improves results over HF-Small (\~456M). Could the authors sat more about where are the diminishing returns? Any evidence that position encodings or window size become the bottleneck next?
> >
> > HF-Large does improve in downstream performance relative to HF-Small, but it also requires substantially longer training. On smaller downstream architectures such as ResNet-18, the gains are consistent but relatively modest which takes into question whether this is an acceptable tradeoff. However, performance differences become more pronounced when sampling weights for larger architectures. This suggests that performance improvements from backbone scaling are downstream architecture-dependent rather than uniform.
> > Regarding bottlenecks, the effective window size of the transformer is a likely constraint. It must be balanced with batch size, memory limits, and backbone scale depending on compute availability. Larger windows are likely beneficial, in particular when processing larger architectures, but at increased computational cost. Furthermore, we are currently using a subset of 2000 models for our experiments and with further backbone scaling this could also be extended to more models or different modalities. As such, while we see improvements from increasing backbone scale, we were not able to exactly determine the point of diminishing returns yet.

---

> ### Author Response · Authors · 2025-11-21
>
> > Zero-shot vs. few-shot. How do results change under N optimizer steps (N∈{0,10,50,100}) with fixed batch size across all architectures? A unified “steps-to-X%” metric would clarify sample-efficiency gains over scratch.
>
> Thank you for the suggestion, this is indeed a valuable metric to assess early convergence gains compared to scratch. Below we show the performance of sampled models on ImageNet in percent (Sec 4.4) averaged per architecture group when using a fixed number of optimization steps with constant batch size 256 as suggested by the reviewer. We will add the detailed results per individual architecture in the revision.
>
> | Architecture | Init/Num Optim Steps    | 0    | 10   | 50    | 100   | 200   | 400   | 500   | 1000  |
> |--------------|----------|------|------|-------|-------|-------|-------|-------|-------|
> | **ConvNeXt** | Sampled  | 0.11 | 4.50 | 20.53 | 22.42 | 25.88 | 35.84 | 45.73 | 57.47 |
> |              | Scratch  | 0.10 | 0.21 | 0.36  | 0.42  | 0.52  | 0.81  | 1.00  | 2.30  |
> | **DenseNet** | Sampled  | 0.26 | 0.18 | 1.09  | 2.95  | 6.49  | 19.57 | 24.40 | 39.69 |
> |              | Scratch  | 0.14 | 0.11 | 0.33  | 0.70  | 1.11  | 1.99  | 3.10  | 5.38  |
> | **EfficientNet**   | Sampled  | 0.17 | 0.11 | 1.94  | 3.45  | 6.56  | 14.91 | 18.89 | 32.33 |
> |              | Scratch  | 0.09 | 0.13 | 0.17  | 0.37  | 0.52  | 1.25  | 1.74  | 4.36  |
> | **MobileNet**| Sampled  | 0.12 | 0.25 | 0.82  | 1.54  | 2.62  | 5.51  | 6.68  | 12.58 |
> |              | Scratch  | 0.10 | 0.10 | 0.33  | 0.52  | 0.94  | 1.88  | 2.38  | 5.18  |
> | **ResNet**   | Sampled  | 1.71 | 6.13 | 16.55 | 23.56 | 28.20 | 37.41 | 42.91 | 55.38 |
> |              | Scratch  | 0.07 | 0.16 | 0.51  | 0.87  | 1.39  | 2.65  | 3.22  | 6.04  |
> | **Swin**     | Sampled  | 0.11 | 0.10 | 0.12  | 0.22  | 0.33  | 0.50  | 0.60  | 2.40  |
> |              | Scratch  | 0.08 | 0.18 | 0.15  | 0.16  | 0.11  | 0.12  | 0.17  | 0.10  |
> | **ViT**      | Sampled  | 0.13 | 0.16 | 0.48  | 0.90  | 1.92  | 3.68  | 4.44  | 9.22  |
> |              | Scratch  | 0.10 | 0.16 | 0.37  | 0.52  | 0.82  | 1.54  | 1.74  | 2.73  |
>
> > For GPT-2, could the authors report exact init metrics (loss/perplexity at step 0) and early convergence deltas vs. scratch across 3 seeds?
>
> Below we report the metrics as requested by the reviewer comparing 3 sampled models to 3 randomly initialized models.
>
> ### Loss
>
> | Model / Num Train Tokens          | 0    | 3,276,800 | 36,044,800 | 68,812,800 | 101,580,800 |
> |-----------------|------|-----------|------------|------------|-------------|
> | **Sampled (AVG)** | **9.7**  | **9.2**       | **8.0**        | **6.4**        | **5.7**         |
> | Model 1         | 9.31 | 8.8       | 7.39       | 6.53       | 6.33        |
> | Model 2         | 9.92 | 9.36      | 8.34       | 6.21       | 5.35        |
> | Model 3         | 10.01| 9.52      | 8.14       | 6.43       | 5.54        |
> | **Scratch (AVG)** | **11.0** | **10.5**      | **9.6**        | **8.7**        | **7.8**         |
> | Model 1         | 10.99| 10.54     | 9.66       | 8.68       | 7.87        |
> | Model 2         | 11.01| 10.48     | 9.56       | 8.67       | 7.6         |
> | Model 3         | 11.0 | 10.52     | 9.63       | 8.68       | 7.89        |
>
> ### Perplexity
>
> | Model / Num Train Tokens    | 0       | 3,276,800 | 36,044,800 | 68,812,800 | 101,580,800 |
> |-----------|----------|-----------|------------|------------|-------------|
> | **Sampled (AVG)** | **18329.1** | **12084.7** | **3990**     | **759**     | **426.7**     |
> | Model 1   | 11435    | 7683      | 1973       | 873        | 716         |
> | Model 2   | 20873 | 13034     | 5464       | 617        | 255         |
> | Model 3   | 22673    | 15537     | 4533       | 787        | 309         |
> | **Scratch (AVG)** | **59765.7** | **38812.7** | **18355.3** | **9016.1**  | **3589.3**    |
> | Model 1   | 59068    | 39905     | 19367      | 9124       | 3915        |
> | Model 2   | 60401    | 37449     | 17204      | 8894       | 2885        |
> | Model 3   | 59828    | 39084     | 18495      | 9030     | 3968        |

---

### Author Response · Authors · 2025-11-21
**Response to Reviewers**

We thank all reviewers for their careful review of the paper and for the detailed, constructive feedback. We appreciate the recognition of the core contributions, including (1) demonstrating that weight-space autoencoders can be trained directly on heterogeneous, uncurated Hugging Face models, (2) the methodological adaptations that make such large-scale heterogeneous training feasible, and (3) the extensive generative evaluation across many architectures and datasets.
Before addressing reviewer-specific questions, we would like to highlight the following:

## Scope of the paper
Our primary goal is to test whether a single backbone can learn a shared latent representation from heterogeneous “in-the-wild” models, and whether this backbone remains functional for common weight-space learning (WSL) downstream tasks as compared to training on curated, laboratory-trained model zoos. We do not aim to optimize every component (e.g., discriminative heads or sampling strategies) nor to claim stronger notions of OOD generalization. Our emphasis is on feasibility and scalability of training on HF models, not on designing new SOTA discriminative or generative methods.


## Interpretation of generative results
Our experiments evaluate whether the learned latent space contains useful structure that can be decoded into workable initializations, following the KDE-guided sampling framework used in previous work [1].  These sampled models are not expected to match the performance of fully trained networks, especially without finetuning, because the weight space backbone is trained across highly diverse architectures and datasets. The observed gains after short finetuning reflect that the decoded weights provide a meaningful starting point rather than a final classifier. We will revise the paper to make this more clear.

More broadly, we would like to reiterate the motivation of the work. Training a weight-space model on uncurated HF weights is challenging, as the data is inherently noisy, heterogeneous, and often undocumented. Our goal is not to claim that this setting is optimal, but to demonstrate that despite this noise, a single backbone can still learn useful and competitive representations given enough training data. This makes WSL significantly more practical: researchers no longer need to train expensive, large-scale model zoos to obtain training data for WSL. We believe lowering this barrier meaningfully broadens access to WSL research and opens the door to applications where curated model zoos are unavailable.

We provide specific responses to each reviewer’s questions and concerns below. We are working on a revision based on the reviewers feedback and will post additional comments with a changelog once it is uploaded.

Thank you again for your time and feedback; it is much appreciated.

---

[1] Schürholt et al., "Towards Scalable and Versatile Weight Space Learning.", ICML 2024.

---

> ### Author Response · Authors · 2025-11-27
> **Revision Changelog**
>
> Here we indicate the changes made to the revision based on the reviewers feedback.
>
> - Extended and clarified the sampling procedure in the Method section (3.1)
> - Provided more details on the evaluation protocol in the Experimental Section (4)
> - Renamed the GPT-2 Experiment section (4.5) to “Generating Weights For Another Modality” and revised the text. Adjusted the conclusion accordingly.
> - Added Appendix D.1.1 where we show the performance of generated models after a fixed number of optimization steps across  architectures with fixed batch size

---

### Meta-Review · Area_Chair_QvBG · 2026-01-06

**Summary:**

The paper shows that high‑quality weight‑space representations can be learned directly from large, heterogeneous model repositories like Hugging Face, outperforming backbones trained on laboratory‑generated model zoos and generalizing to unseen modalities. Most reviewers find that the paper targets an important and emerging task, and the idea of learning from heterogeneous model collections interesting.

**Reviewer Concerns:**

Several major concerns were raised, including the need for a clearer definition of a single weight‑space representation, more details on experimental design and evaluation, limited diversity of experiments, the claim of generalization to new modalities or OOD tasks, the influence of the KDE bandwidth, and differences from SANE. The authors provided brief responses to most of these points, and the AC finds that some concerns have been only partially addressed.

**Reviewer Scores:**

This paper receives the following ratings: Marginally Above, Marginally Below, and Reject. If the reviewers had been able to participate fully in the discussion, the AC would expect negative ratings to remain, as some concerns were only partially addressed. The AC recommends not accepting the paper in its present form.

---

### Decision · Program_Chairs · 2026-01-26

Reject